# DoReMi - Difficulty-Oriented Reasoning Effort Modeling of Science Problems for Large Language Models

## Abstract

We introduce DoReMi (Difficulty-Oriented Reasoning Effort Modeling), a structured framework leveraging an extended Bloom's taxonomy to comprehensively characterize intrinsic problem difficulty for large language models on scientific reasoning tasks. DoReMi systematically annotates problems along six cognitive and methodological axes using judge large language models (LLM) distinct from those being evaluated, with human annotations confirming the validity of these assessments. We empirically quantify LLM reasoning effort through metrics including minimum reasoning tokens required for solution, expected number of attempted runs to first correct answer. Our validation demonstrates strong agreement across diverse judge LLMs spanning both open-source and proprietary LLMs. Evaluations on GPQA, ARC, and SuperGPQA reveal that our multidimensional difficulty fingerprints correlate strongly with and enable accurate predictive modeling of LLM reasoning effort. DoReMi enables principled difficulty-aware subset selection that substantially outperforms other baselines while providing interpretable diagnostics that uncover emergent reasoning capabilities across successive model generations. This framework offers actionable insights for benchmark design and targeted post-training improvements toward higher-order reasoning skills.

## 1 Introduction

Latest reasoning large language models (LLMs) have demonstrated significant progress in tackling complex reasoning tasks. However, clearly characterizing their capabilities remains challenging, as task difficulty often combines several partially overlapping factors including linguistic complexity, domain-specific knowledge, and the depth of reasoning involved. Consider a partial differential equation problem: an LLM might present a correct solution either by recalling a known theorem or by logically deriving the solution from fundamental principles. Both approaches are expressed through language and rely on familiarity with domain-specific notation, such as distinguishing between $\mathbf{x}$ and $\vec{x}$. When conventional benchmarks fold these heterogeneous challenges into a single accuracy score, they hide which capability—domain knowledge, deductive reasoning, or methodological complexity—were the real bottlenecks. Even seemingly finer-grained signals, like the accuracies of a question across a leaderboard packed with hundreds of LLMs, tell us little: most leaderboard entries are generic non-reasoning models, so their collective failure is like asking a roomful of laypeople to solve a PhD-level physics question—the near-universal miss reflects the respondents more than the task. Without a nuanced and principled way to measure difficulty for reasoning LLMs, tracking progress across model versions becomes inconsistent. New benchmarks risk being quickly saturated by state-of-the-art LLMs. Applications like curriculum learning or difficulty-aware subset sampling, which depend on understanding why a question is hard, remain mostly heuristic.

To address these limitations, we propose a *structured, multi-dimensional* evaluation framework grounded in educational theory, specifically Bloom's taxonomy and its extensions Heer (2012). Our framework systematically annotates each scientific reasoning problem along six complementary axes: *Cognitive Level*, *Knowledge Dimension*, *Method Difficulty*, *Definition Completeness*, *Knowledge Breadth*, and *Number of Reasoning Steps*. Although not strictly orthogonal, these dimensions provide principled, theory-informed handles that expose facets of difficulty invisible to traditional static metrics such as SMOG, Gunning Fog, and Flesch-Kincaid scores McLaughlin (1969); Scott (2025);

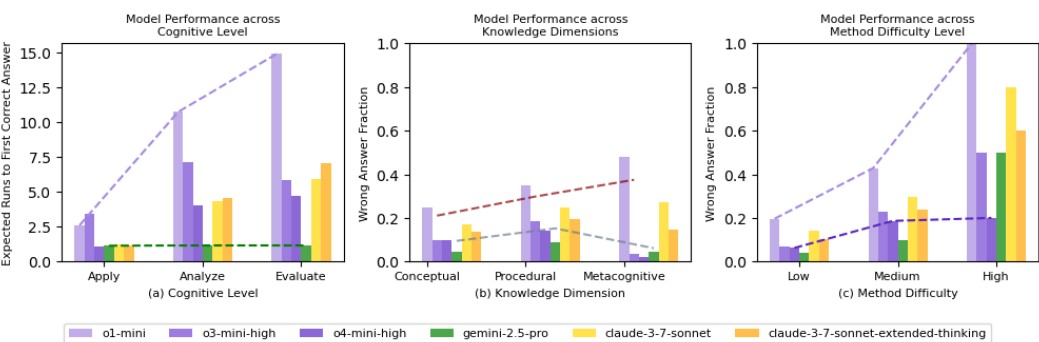

Figure 1: The performance of different LLMs long three Bloom axes: (a) cognitive level, (b) knowledge dimension, and (c) method difficulty.

Tanprasert & Kauchak (2021). The resulting interpretable "difficulty fingerprint" explicitly captures both the *what* (knowledge and cognition) and the *how* (methodological effort) necessary to reach a solution. This approach supports more nuanced performance comparisons across LLMs and helps evaluate and improve the difficulty of both current and future benchmarks.

One key application of this framework is to reveal unique insights into how intrinsic problem characteristics influence the reasoning effort required by LLMs for scientific problem-solving. Figure 1 highlights three important trends. **(i) Cognitive Level:** Early-generation reasoning models like `o1-mini` exhibit a steep increase in reasoning effort (measured by the expected number of runs to hit the first correct answer) as cognitive complexity increases, whereas advanced current-generation models such as `gemini-2.5-pro` demonstrate remarkable stability, maintaining consistently low reasoning effort across cognitive levels. **(ii) Knowledge Dimension:** The first-generation reasoning model `o1-mini` and the non-reasoning mode of `claude-3.7-sonnet` show a clear rise in error rates as problems require higher-order knowledge dimensions. In contrast, reasoning enabled (`claude-3.7-sonnet-extended-thinking`) and other later-generation reasoning LLMs exhibit robust, stable performance or even improved accuracy at the highest (metacognitive) dimension, suggesting emergent capabilities along the knowledge dimension axis. **(iii) Method Difficulty:** we observe improved robustness across successive generations **in the same model family** as `o1-mini` shows a sharp rise in errors from *Low* to *High* methodological difficulty, while `o3-mini-high` exhibits a more gradual increase and `o4-mini-high` maintains stable performance from *Medium* to *High*. These diagnostic insights reveal a gradual and dimension-specific emergence of higher-order reasoning skills. This pattern supports our hypothesis that scaled reinforcement learning (RL) during post-training may lead to effective transfer of reasoning strategies. It also enables stronger cognitive generalization across the Bloom-inspired axes. Our paper contributes three primary advancements:

1. **Difficulty-Oriented Reasoning Effort Modeling (DoReMi).** We introduce a framework to model difficulty for reasoning-intensive science problems. By correlating structured difficulty metrics derived from an extended version of Bloom's taxonomy with empirically measurable proxies of *reasoning effort*, such as the *minimum reasoning token length (MRT)* and *expected number of attempted runs to first correct answer (R2FCA)*, we establish interpretable, dimension-specific relationships between intrinsic problem characteristics and LLM reasoning demands. Based on these insights, we develop and validate predictive models capable of accurately identifying challenging ("high-effort") problems across multiple established scientific reasoning benchmarks such as GPQA Rein et al. (2024), ARC Clark et al. (2018), and SuperGPQA Team et al. (2025).

2. **Difficulty-Aware Downstream Applications.** Using the predictive capabilities of our DoReMi framework, we propose difficulty-aware subset selection, which adaptively prioritizes challenging problems. This helps preserve discriminative evaluation power even as aggregate benchmark performance saturates due to rapid advances in LLM capabilities. Experiments comparing our approach against static-difficulty baselines show significant improvements in correctly identifying truly challenging problems. Furthermore, these difficulty-aware selection strategies could enable more informed curriculum learning protocols and facilitate targeted benchmark refinement.

3. **Interpretable Diagnostics of Reasoning Capabilities.** We leverage the fine-grained difficulty characterizations provided by DoReMi to construct a diagnostic framework for systematically analyzing reasoning strengths and weaknesses of LLMs. By examining performance stratified along multiple axes of Bloom's taxonomy, we uncover distinct patterns in reasoning behaviors

across generations of models and post-training stages. Our analyses pinpoint precisely how and where improvements manifest—highlighting emergent capabilities especially along cognitive and knowledge dimension axes. These gained insights could provide potential guidance on post-training methodologies to foster higher-order reasoning skills.

## 2 DESIGN PRINCIPLES

There is currently a lack of precise and reliable measures of intrinsic problem difficulty tailored specifically to reasoning capabilities. Traditional static metrics, such as readability scores, overlook cognitive and methodological complexity inherent to reasoning problems.

### 2.1 ANALOGY AND INTUITION: REASONING EFFORT AS A COGNITIVE BUDGET

We propose to operationalize problem difficulty through explicit, measurable proxies for *reasoning effort*, directly mirroring cognitive processes observed in human scientific problem-solving:

**Expected Number of Attempted Runs to First Correct Answer (R2FCA)** Analogous to a mathematician discarding one proof sketch after another until a promising idea emerges, we measure the average number of independent attempts an LLM needs before it first produces the correct answer. This metric reflects both the model's exploratory persistence and its inherent stochasticity.

**Minimum Reasoning Token (MRT)** required to solve a problem from multiple sampled solutions, retaining only the shortest successful one. This is the LLM analogue of a mathematician's cleanest proof, capturing the minimal cognitive and computational budget required to solve the problem.

Consider the cognitive process a mathematician undergoes when confronting a challenging theorem. Typically, they begin by exploring various potential proof strategies—each attempt consuming significant time, effort, and cognitive resources analogous to iterative "scratch paper" explorations. As many explored pathways fail to yield immediate success, repeated exploratory iterations often become necessary. Thus, if the probability of reaching a correct solution per attempt is low, the expected number of exploratory attempts scales inversely with this probability. Additionally, even after identifying a viable strategy, the final formal proof still incurs an inherent cognitive "cost"—the minimal sequence of logical reasoning steps (or written tokens) required to rigorously articulate the solution. In this analogy, our notion of *Expected Reasoning Cost (ERC)* aligns with:

$$\text{Expected Reasoning Cost (ERC)} \sim \underbrace{E[\text{R2FCA}]}_{\text{Exploratory Attempts}} \otimes \underbrace{\text{MRT}}_{\text{Minimal Solution Transcript}}$$

as a combined metric indicated by symbolic $\otimes$ that jointly captures the iterative exploratory complexity and the minimal cognitive burden to reach a successful solution. Applying this to LLM reasoning under stochastic sampling, the repeated attempts are like the mathematician's exploration, and the generated reasoning tokens represent the cognitive resources used.

### 2.2 EXTENDED BLOOM'S TAXONOMY

To capture the multifaceted nature of science problem, we extend the original Bloom's framework into a *six–axis taxonomy*. Each axis is annotated directly from the problem statement and reference solution, yielding a machine-parsable "difficulty fingerprint".

1. **Cognitive Level**: Highest Bloom cognitive process required to solve the task: *Remember*, *Understand*, *Apply*, *Analyze*, *Evaluate*, or *Create*. The level is chosen by locating the most demanding mental operation that a correct solution must exhibit.
2. **Knowledge Dimension**: Type of knowledge invoked: *Factual*, *Conceptual*, *Procedural*, or *Metacognitive*. This axis distinguishes mere recall from methodological know-how and self-regulation of the reasoning process.
3. **Method Difficulty**: Degree of methodological novelty (*Low*, *Medium*, *High*). *Low* denotes routine, textbook procedures; *Medium* requires minor adaptations or synthesis; *High* entails non-routine combinations or inventive leaps.
4. **Definition Completeness**: Whether the statement fully specifies the solution space (*Complete*) or leaves essential variables/criteria implicit (*Incomplete*), forcing the solver to supply assumptions.

5. **Knowledge Breadth** Disciplinary span of required knowledge: *Single-* versus *Multi-Discipline*. The latter flags problems that integrate concepts from two or more distinct scientific fields.
6. **Number of Reasoning Steps**: Integer count of essential logical actions whose removal would break the solution chain. Trivial paraphrases are excluded.

Together, these six axes disentangle the *what* (knowledge and cognition) from the *how* (method and reasoning) of problem solving, enabling the reasoning effort modeling based on these metrics.

## 3 RELATED WORK

Recent advancements in reasoning LLMs have highlighted the need for precise measures of problem difficulty, specifically within scientific tasks where reasoning complexity is prominent. Traditional assessments have relied on static readability metrics, such as SMOG, Gunning Fog, or Flesch-Kincaid McLaughlin (1969); Scott (2025); Tanprasert & Kauchak (2021), which fail to reflect deeper cognitive demands required in complex reasoning tasks. Prompt-based approaches Rooein et al. (2024) partially overcome these limitations by leveraging LLMs' language understanding capabilities to capture more abstract complexity; however, they primarily emphasize textual difficulty rather than cognitive or methodological complexity. Compared to these static readability-focused metrics, our DoReMi framework leverages an extended Bloom's taxonomy Heer (2012), systematically characterizing multiple dimensions of intrinsic cognitive and methodological difficulty.

Previous studies have employed Bloom's taxonomy in the context of LLM evaluation Huber & Niklaus (2025) and curriculum learning design Hase et al. (2024). Huber et al. Huber & Niklaus (2025) classified benchmarks according to Bloom's cognitive levels, revealing that LLM performance predominantly excels at lower cognitive levels. Complementing this work, DoReMi enriches Bloom's taxonomy with additional methodological and metacognitive axes, directly linking these theory-informed dimensions with empirical proxies of reasoning effort rather than accuracy alone.

Curriculum learning shows performance gains when aligning difficulty progression for LLM training. To quantify sample-level hardness, previous work Hase et al. (2024) annotated the Bloom's taxonomy for each dataset based solely on human annotation. In contrast, we refined annotation with multiple LLM judges and validated them with human annotations to ensure scalability and consistency.

Recent research investigates the relation between chain-of-thought (CoT) lengths and reasoning success, suggesting a non-monotonic relationship and an optimal length dependent on problem complexity and model capacity Wu et al. (2025). In comparison to this previous analysis, DoReMi explicitly correlates intrinsic problem properties to proxies of reasoning effort, modeling and predicting effective reasoning difficulties across multiple science benchmarks.

## 4 SOLUTION

In this section, we describe our systematic methodology to predict LLM reasoning effort by combining difficulty metrics from an extended 3D Bloom's Taxonomy with learned reasoning patterns.

### 4.1 PROBLEM DIFFICULTY QUANTIFICATION VIA 3D BLOOM'S TAXONOMY

We develop a systematic annotation process using our extended 3D Bloom's taxonomy (Section 2.2) to quantify problem difficulty across $D = 6$ dimensions: *Cognitive Level*, *Knowledge Dimension*, *Method Difficulty*, *Definition Completeness*, *Reasoning Steps*, *Knowlege Breath*.

For each problem, we employ $K$ reasoning LLMs as automated judges. Each judge independently evaluates the problem statement and reference solution, classifying difficulty along each Bloom dimension with supporting rationale. Through iterative prompt refinement—testing over two dozen variants with explicit decision criteria and illustrative examples—we achieved strong inter-judge agreement. The optimized prompts are provided in Section 7.2. We also validated our approach by comparing average LLM-as-a-judge scores against two human annotations on 100 GPQA questions, with good alignment results observed in Section 5.1.

Final annotations are obtained by encoding categorical Bloom levels into ordinal scores $b_{i,d} \in \{1, 2, ..., L_d\}$ for dimension $d$ and problem $i$, then averaging across judges. This yields a continuous multi-dimensional "difficulty fingerprint" $B_i = \{b_{i,d}\}_{d=1}^D$ for each problem.

### 4.2 REASONING EFFORT METRICS

To capture the computational and algorithmic effort required for reasoning, we evaluate $M$ reasoning models, each for $R_m$ runs per question. We define four complementary effort metrics:

**(1) Wrong Answer Fraction (WAF):** The average failure rate across models:

$$\text{WAF}_i = \frac{1}{M} \sum_{m=1}^M \left( \frac{1}{R_m} \sum_{r=1}^{R_m} \mathbf{1}(s_{i,m,r} = 0) \right)$$

where $s_{i,m,r} \in \{0, 1\}$ indicates correctness for question $i$, model $m$, run $r$.

**(2) Minimum Reasoning Tokens (MRT):** The minimum token count needed for success:

$$\text{MRT}_{i,m} = \begin{cases} \min\{t_{i,m,r} \mid s_{i,m,r} = 1\}, & \text{if any run succeeds} \\ \max\{t_{i,m,r} \mid r \in [R_m]\}, & \text{otherwise} \end{cases}$$

where $t_{i,m,r}$ denotes reasoning tokens used.

**(3) Expected Runs to First Correct Answer (R2FCA):** The expected number of attempts needed:

$$\text{R2FCA}_i = \sum_{n=1}^{R_m} n \cdot P(n) + \epsilon$$

where $P(n)$ is the empirical probability of first success on run $n$. Taking the expectation across many trials smooths out single–run volatility; the $\varepsilon$ floor prevents division by 0 when no run succeeds. R2FCA is intended to measure *problem solvability under repeated attempts*.

**(4) Reasoning Inconsistency (RI):** The diversity of reasoning trajectories:

$$\text{RI}_i = \frac{1}{R_m} \sum_{r=1}^{R_m} d(c, e_r), \quad c = \frac{1}{R_m} \sum_{r=1}^{R_m} e_r$$

where $e_r$ are response embeddings, $c$ is their centroid, and $d(\cdot, \cdot)$ is cosine distance.

### 4.3 LEARNING TO PREDICT REASONING EFFORT

Our correlation analysis in Section 5.2 reveals that MRT exhibits the strongest relationship with Bloom metrics. Based on this finding, we develop a two-stage approach to predict reasoning effort:

**Stage 1: Model Aggregation.** We combine model-specific MRTs into a unified metric:

$$\text{MRT}_C = \sum_{m=1}^M w_m \cdot \text{MRT}_{i,m}$$

where weights $w_m$ are learned via gradient descent to maximize F1-score on high-effort samples. $MRT_C$ is distributed in four quantile-based bins — Minimum, Low, Medium, and High—ensuring clear separation of difficult problems from the abundant easy ones. These binned labels serve as targets/ground truth for training our reasoning-effort predictor.

$$\text{Category}(x) = \begin{cases} \text{Minimum}, & x < \mu - \sigma \\ \text{Low}, & \mu - \sigma \leq x < \mu \\ \text{Medium}, & \mu \leq x < \mu + \sigma \\ \text{High}, & x \geq \mu + \sigma \end{cases}$$

**Stage 2: Difficulty-to-Effort Mapping.** We train a neural classifier $f_\theta : B_i \to \text{Category}$ that maps Bloom features to effort categories. The model is optimized using weighted cross-entropy loss with

emphasis on high-effort samples:

$$\mathcal{L} = -\sum_{i=1}^{N} \sum_{c=1}^{4} \alpha_c \cdot y_{i,c} \log(f_\theta(B_i)_c)$$

where $\alpha_c$ are class weights and $y_{i,c}$ are one-hot encoded labels.

### 4.4 DoReMi Algorithm

Algorithm 1 summarizes our complete pipeline: (1) annotate bloom metrics using LLM judges, (2) inference of reasoning LLMs to collect reasoning effort metrics, (3) learn optimal aggregation weights, and (4) train a predictor from difficulty features to effort categories.

---

**Algorithm 1** DoReMi: Difficulty-oriented Reasoning Effort Modeling

---

**Require:** Questions $Q = \{q_i\}_{i=1}^{N}$, Models $M = \{m_j\}_{j=1}^{M}$, Judges $J = \{j_k\}_{k=1}^{K}$
**Ensure:** Effort predictor $f_\theta : B \rightarrow$ Category
 1: **// Phase 1: Difficulty Annotation**
 2: **for** each question $q_i \in Q$ **do**
 3:     Obtain Bloom features $B_i$ via judge consensus (Section 3.1)
 4: **end for**
 5: **// Phase 2: Effort Measurement**
 6: **for** each model $m_j \in M$, question $q_i \in Q$ **do**
 7:     Evaluate $R_j$ runs, record correctness and token counts
 8:     Compute $\text{MRT}_{i,j}$ using Equation (2)
 9: **end for**
10: **// Phase 3: Learn Aggregation**
11: Initialize weights $w = [w_1, ..., w_M]$ randomly
12: **while** not converged **do**
13:     $\text{MRT}_C \leftarrow \sum_j w_j \cdot \text{MRT}_{i,j}$
14:     Update $w$ to maximize F1 on high-effort class
15: **end while**
16: **// Phase 4: Train Predictor**
17: Discretize $\{\text{MRT}_{C,i}\}$ into categories
18: Train $f_\theta$ on dataset $\{(B_i, \text{Category}_i)\}_{i=1}^{N}$
19: **return** Trained predictor $f_\theta$

---

## 5 Experiments and Results

Comprehensive experiments were conducted to validate our DoReMi framework. We evaluated 9 LLMs: three generations of OpenAI reasoning models (`o1-mini`, `o3-mini-high`, `o4-mini-high`); three generations of Qwen-14B model family including two non-reasoning variants (`Qwen1.5-14B`, `Qwen2.5-14B`) plus one reasoning variant (`Qwen3-14B`); Google's `gemini-2.5-pro`; and Anthropic's `claude-3.7-sonnet` (non-reasoning mode) and `claude-3.7-sonnet-extended-thinking` (reasoning mode). Each model answers every problem 10 times to capture stochasticity, yielding 90 runs per problem.

Our evaluation spans three scientific reasoning benchmarks representing varying difficulty distributions: GPQA (PhD-level STEM), ARC (K-12 science reasoning), and two specialized SuperGPQA domains (particle physics and molecular biology) test deep domain expertise.

### 5.1 Bloom Scores: Alignment between judge groups

To validate our automated Bloom annotations, we compared LLM-generated scores with human judgments on 100 GPQA problems. Two independent human annotators rated all six Bloom dimensions; Human1 served as the reference. We used three open-source reasoning LLMs—`DeepSeek-V3.1-Terminus` DeepSeek-AI (2024), `GPT-OSS-120B` OpenAI (2025), `Qwen3-Next-80B-A3B-Thinking` Yang et al. (2025)—to avoid overlap with our

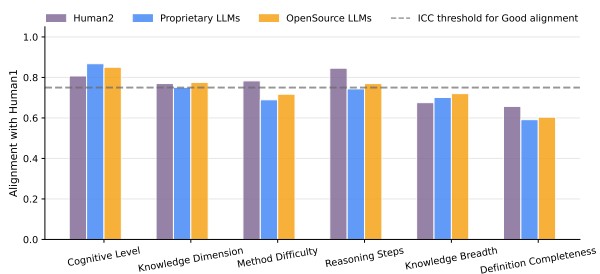
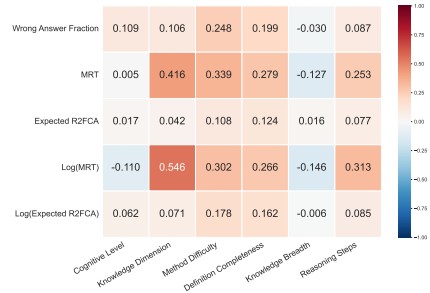

Figure 2: Human–model alignment across Bloom metrics. ICC(2,1) values measure inter-rater reliability between Human1 (reference) and three annotator groups.

Figure 3: Correlation Analysis of Reasoning Effort proxies across Bloom Axes

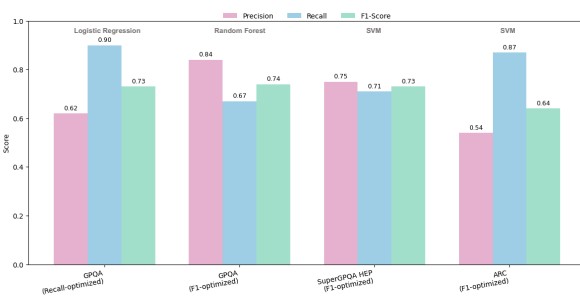
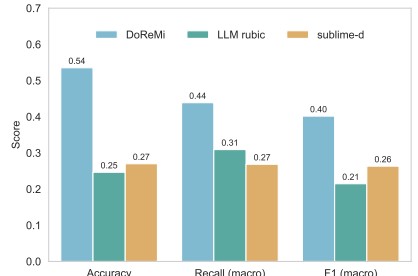

Figure 4: Test performance of optimal classification model for high-effort prediction on GPQA, SuperGPQA-HEP, and ARC using recall and f1-score optimization

Figure 5: Benchmarking DoReMi with baselines in Predicting Reasoning Effort

evaluation models, and three proprietary LLMs—`gemini-2.5-pro`, `gemini-2.5-flash`, `o3-mini`—to test cross-group consistency.

Figure 2 reports Interclass Correlation Coefficient Leyland & Groenewegen (2014) - ICC(2,1) across judge groups. LLM judges show good–to–excellent agreement with humans on the axes most predictive of effort—*Cognitive Level*, *Knowledge Dimension*, and *Reasoning Steps*—with ICC > 0.75, and approach inter-human reliability on most remaining axes. Open-source and proprietary judges align closely with each other (Appendix Fig. 11).

Bloom-axis judging is rubric-driven, simpler than verifying full solutions, and stable across judge groups, mitigating LLM-as-judge risks. We therefore use LLM-derived Bloom features in DoReMi.

## 5.2 REASONING EFFORT MODELING RESULTS

We evaluated candidate effort proxies—WAF, R2FCA, UCA, and RI—and analyzed their correlations with Bloom axes (Fig. 3). MRT emerged as the strongest signal, so we adopt it as the primary proxy and aggregate it across $M$ reasoning models into a combined metric $MRT_c$. Model-specific weights (Figs. 24, 25) are learned to optimize either recall- or $F_1$-oriented objectives. The resulting aggregate achieves $\approx 80\%$ precision for identifying high-effort problems (details later). While a future composite could integrate R2FCA, WAF, and related proxies, the $MRT_c$-based approach is a strong, practical baseline.

We train classifiers to detect high-effort questions (high $MRT_c$). Figure 4 summarizes precision–recall trade-offs under recall- and $F_1$-optimized settings. On **GPQA**, optimizing for recall selects logistic regression (recall 0.90, precision 0.62); optimizing for macro-$F_1$ favors a linear SVM (precision 0.81, recall 0.71). Five-fold CV yields $F_1 = 63.4\% \pm 4.9\%$, precision $57.2\% \pm 2.4\%$, and recall $71.5\% \pm 9.3\%$ (Fig. 27). Our evaluation spans both science-focused benchmarks (GPQA, SuperGPQA-HEP) and general reasoning tasks (ARC) to demonstrate that Bloom taxonomy metrics can effectively capture reasoning difficulty across different domains. On **SuperGPQA-HEP**, SVM attains balanced performance (precision 75%, recall 71%). On **ARC**, SVM prioritizes recall

(87%) at lower precision (54%). Across benchmarks, the models remain interpretable and require no dataset-specific feature engineering, further supporting the domain-agnostic utility of Bloom taxonomy features for reasoning effort prediction.

### 5.3 DOReMI USE CASE 1: DIFFICULTY-AWARE SUBSET SELECTION

As state-of-the-art LLMs achieve near-saturation performance on many benchmarks (e.g., 85-88% on GPQA), distinguishing between models becomes increasingly challenging. Difficulty-aware subset selection addresses this by strategically sampling challenging problems to create more discriminative evaluation sets. We compare three approaches for identifying high-effort problems:

**DoReMi (Ours):** Leverages learned reasoning effort models based on 3D Bloom taxonomy metrics to predict $M\hat{R}T_c$, capturing nuanced aspects of problem difficulty beyond surface-level complexity. The configurable effort metric $MRT_c^*$ allows optimization for different objectives (recall vs. F1), enabling flexible prioritization of either coverage or precision in identifying high-effort questions.

**Sublime-D Baseline:** A specialized variant within the SubLIME framework Xu et al. (2024) that uses static readability metrics (Flesch, Gunning Fog) to estimate difficulty. While computationally efficient, these surface-level metrics may miss deeper reasoning challenges that make problems truly difficult for LLMs.

**LLM Rubric Baseline:** Employs a judge LLM (o4-mini-high) to directly classify reasoning effort by analyzing problem statements and reference solutions holistically. This end-to-end approach serves as a methodological contrast to DoReMi's structured, multi-dimensional analysis (full prompt in Appendix 7.6.1).

Figure 5 demonstrates DoReMi's superior performance in identifying genuinely challenging problems. DoReMi achieves 54% accuracy in categorizing high reasoning effort problems, significantly outperforming both the LLM rubric (25%) and Sublime-D (27%) methods. This 2× improvement suggests that structured analysis through Bloom taxonomy features provides more reliable difficulty assessment than either static readability metrics or direct LLM classification.

The practical impact is substantial: when selecting a discriminative subset from benchmarks, DoReMi's higher accuracy ensures more "real" hard samples are included, preserving evaluation power even as aggregate performance saturates. This capability enables more informed curriculum learning protocols and facilitates targeted benchmark refinement as LLM capabilities continue to advance.

### 5.4 DOReMI USE CASE 2 - INTERPRETABLE DIAGNOSTICS OF REASONING CAPABILITY

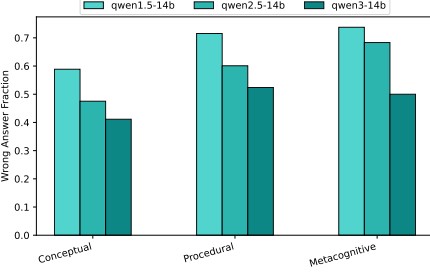

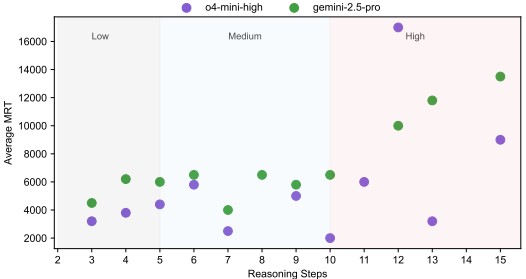

Figure 6: Qwen 14B model family analysis: Wrong answer fractions along knowledge-dimension axis

Figure 7: Average MRT vs Reasioning steps

**Cognitive Level.** Fig 1a shows the *expected number of R2FCA*, grouped by Bloom's cognitive levels. For `o1-mini`, the curve rises sharply—roughly quadrupling from APPLY to ANALYZE, and again from ANALYZE to EVALUATE. Since this metric is inversely related to a model's per-attempt success rate, the steep slope shows that early-generation reasoning models experience an exponential drop in hit rate as soon as multi-step evaluation or hypothesis testing is needed. In contrast, `gemini-2.5-pro` maintains an almost flat profile. Its expected runs change little across the three

cognitive tiers, suggesting both a higher baseline competence and a much stronger ability to transfer reasoning strategies as the cognitive level of the science problem increases. `o3-mini-high` and `o4-mini-high` fall between these extremes. This suggests a smooth, but not yet complete, scaling path where each new model narrows the gap between APPLY-level heuristics and EVALUATE-level analytical reasoning.

**Knowledge Dimension.** Fig 1b reports the *wrong–answer fraction* (WAF), grouped by Bloom's knowledge dimensions. `o1-mini` again displays a monotonically rising error profile: its WAF climbs from the CONCEPTUAL band through PROCEDURAL and peaks at the METACOGNITIVE tier. Scaling post-training in `o3-mini-high` and `o4-mini-high` eliminate the gap between CONCEPTUAL and PROCEDURAL questions and even invert the trend at METACOGNITIVE. A similar transition is found in Claude: without reasoning tokens, performance of `claude-3.7-sonnet` tracks `o1-mini` almost exactly—error rates rise steadily with the knowledge dimension, suggesting that the core model alone shares the same weakness. When reasoning tokens are enabled in `claude-3.7-sonnet-extended-thinking`, the pattern breaks: errors grow only slightly from CONCEPTUAL to PROCEDURAL and *decrease* at METACOGNITIVE. The open-source Qwen-14B family shows the same story (Fig. 6): `Qwen1.5-14B` and `Qwen2.5-14B`—which lack RL-based post-training—exhibit a monotonic rise from CONCEPTUAL to METACOGNITIVE, whereas the reasoning-tuned `Qwen3-14B` reverses the trend with lower error at METACOGNITIVE than PROCEDURAL. Because RL appears only in the Qwen 2.5→3 transition, these results provide converging evidence that RL might be the driver of the emergent metacognitive capability.

**Method Difficulty.** Fig 1c reveals a clear stratification in how successive model generations respond to increasing *method difficulty*. For `o1-mini`, the wrong–answer fraction rises sharply—from roughly $20\%$ at *Low* to $42\%$ at *Medium*, and up to $100\%$ at *High*. `o3-mini-high` exhibits the same monotonic trend but with a noticeably shallower slope. In contrast, `o4-mini-high` shows a qualitatively different pattern: its error rate increases from *Low* to *Medium* and then *plateaus*, remaining statistically unchanged from *Medium* to *High*, which indicates a newfound robustness. Compared to the knowledge-dimension results, these observations point to a *staggered emergence* of higher-order abilities: sensitivity to *what* knowledge is required appears earlier (in `o3-mini-high`), while resilience to *how* that knowledge must be operationalized emerges one generation later (in `o4-mini-high`). This suggests that scaled RL post-training yields non-uniform gains across difficulty axes, with different dimensions reaching their inflection points at distinct stages.

**Solution-Step Complexity vs. Minimum Reasoning Tokens**. Figure 7 reveals a *piecewise* relationship between the annotated number of solution steps and the *minimum* reasoning tokens required by the most capable models in our study, `o4-mini-high` (•) and `gemini-2.5-pro` (•). In the **low-complexity regime** (steps $< 5$), both models quickly converge on concise answers—the average reasoning token *falls* as the step count decreases. Once problems demand **more than 10 explicit steps**, each additional step now incurs a significant rise in the minimum token count. This suggests that beyond a critical complexity threshold the models must maintain longer context windows to keep intermediate facts "alive," and the cost scales super-linearly with step count. Intriguingly, in the **transition band** ($5 \leq$ steps $\leq 10$) the correlation is almost flat for both systems. Together, these three regimes highlight an emergent efficiency plateau followed by an exponential token explosion.

## 6  CONCLUSION

We proposed DoReMi, a structured Bloom-inspired framework for quantifying science problem difficulty for LLMs. Evaluations on GPQA, ARC, and SuperGPQA demonstrated good correlations (up to (r=0.52)) between our multidimensional difficulty fingerprints and empirical reasoning effort metrics MRT and expected R2FCA. Our reasoning-effort prediction model significantly outperformed static-difficulty baselines: DoReMi (54%) vs Sublime-D (27%). Moreover, through interpretable diagnostics of reasoning capabilities, we identified emergent reasoning capabilities—such as improved robustness to methodologically challenging problems and enhanced metacognitive monitoring—that have appeared systematically across successive generations of reasoning LLMs. These insights could potentially guide future targeted model improvements, curriculum design, and benchmark creation, particularly highlighting the importance of careful post-training strategies for fostering higher-order reasoning capabilities.

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

# 7 TECHNICAL APPENDICES AND SUPPLEMENTARY MATERIAL

## 7.1 THE USE OF LLMs IN MANUSCRIPT PREPARATION

In accordance with ICLR 2026 submission guidelines, we disclose the use of Large Language Models (LLMs) in the preparation of this manuscript. LLMs were employed as general-purpose assist tools for the following specific purposes:

- **Writing assistance**: LLMs were used to improve clarity, grammar, and flow of certain sections of the manuscript, particularly in refining technical explanations and ensuring consistent terminology throughout the paper.
- **Literature review support**: LLMs assisted in identifying relevant related work and helped structure the presentation of background material, though all cited works were independently verified by the authors.
- **Code documentation**: LLMs were used to generate comments and documentation for supplementary code materials to improve readability and reproducibility.

We emphasize that the core research ideas, experimental design, methodology, analysis, and conclusions presented in this work are entirely the intellectual contribution of the human authors. LLMs did not play a significant role in research ideation or the formulation of novel contributions. All factual claims, experimental results, and scientific interpretations have been independently verified by the authors.

The authors take full responsibility for all content in this manuscript, including any text that may have been refined with LLM assistance. We confirm that no content generated by LLMs could be construed as plagiarism or scientific misconduct, and all sources and prior work are properly attributed.

## 7.2 FULL BLOOM'S TAXONOMY PROMPT TEMPLATES

This appendix provides the full prompt templates used for automated, LLM-based annotation of science problems along the extended 3D Bloom's taxonomy. Each template is optimized for consistency

and clarity, facilitating accurate evaluation by large language models. Prompts are presented verbatim as used in our experiments.

PROMPT 1: COGNITIVE LEVEL ASSESSMENT

```
You are an expert in evaluating scientific problems using Bloom's
    Taxonomy. Your task is to assess a given problem statement and its
    reference solution to determine the highest cognitive process level
    required for its resolution. Focus solely on the cognitive actions
    necessary to solve the problem. Your analysis should be objective,
    detailed, and applicable across all scientific subdomains.

The cognitive process levels (from lowest to highest) are defined as
    follows:

1.  Remember : Involves the recall or retrieval of factual information,
    definitions, or previously learned material without modification or
    interpretation.
    *Example tasks:* Listing key facts, reciting definitions, or recalling
     formulas.

2.  Understand : Involves demonstrating comprehension by interpreting,
    summarizing, or explaining concepts in your own words.
    *Example tasks:* Paraphrasing theories, summarizing research findings,
     or explaining the meaning of concepts.

3.  Apply : Involves using known information, methods, or procedures in
    specific or novel situations.
    *Example tasks:* Solving standard problems using known formulas,
    applying theories to new contexts, or executing established
    procedures.

4.  Analyze : Involves breaking complex information into parts to examine
     relationships, identify patterns, and differentiate between
    components.
    *Example tasks:* Decomposing arguments, comparing and contrasting
    elements, or mapping relationships within a system.

5.  Evaluate : Involves making judgments based on criteria or standards
    by critiquing, assessing, or validating theories, methods, or
    evidence.
    *Example tasks:* Critically assessing the validity of a hypothesis,
    weighing evidence, or comparing alternative approaches.

6.  Create : Involves synthesizing information to produce new or original
     ideas, models, or solutions.
    *Example tasks:* Formulating novel hypotheses, designing innovative
    experiments, or constructing comprehensive models that integrate
    diverse elements.

===================================================
Problem Statement:
<problem_statement>
{problem_statement}
</problem_statement>

===================================================
Reference Solution:
<reference_solution>
{reference_solution if reference_solution is not None else  None }
</reference_solution>

Instructions:
```

```
a. Read the problem statement and reference solution carefully.
b. Identify the specific cognitive actions required for a correct
    solution.
c. Determine the highest cognitive process level (from the list above)
    necessary to solve the problem.
d. Provide a detailed rationale for your rating by citing specific
    elements from the problem statement and reference solution that
    indicate the required cognitive processes.

Now output your answer as a JSON object with two keys: cognitive_level
    (which must be one of Remember, Understand, Apply, Analyze,
    Evaluate, or Create) and rationale. Do not include any
    additional commentary.
```

PROMPT 2: METHOD DIFFICULTY ASSESSMENT

```
You are an expert in evaluating scientific problems using Bloom's 3D
    Taxonomy.

**Objective:**
Evaluate the difficulty of the method required to solve a problem,
    focusing exclusively on the methods characteristics. Base your
    judgment solely on the method's nature-not on the overall complexity
    of the scientific or biological context.

**Classification Criteria:**

- **Low Method Difficulty:**
  - The problem is solved by directly applying a well-known, established
    method exactly as described in standard references.
  - No modifications, adaptations, or additional interpretative steps are
     necessary.

- **Medium Method Difficulty:**
  - The solution requires adapting or modifying a known method to fit the
     specific problem constraints or context.
  - This may include reconciling contradictory observations, integrating
    multiple pieces of evidence, or ruling out alternative hypotheses
    before arriving at a conclusion.
  - *Threshold:* If even a single additional step beyond the basic method
     is needed (e.g., adjusting for indirect binding signals or combining
     two standard analyses), classify as Medium.

- **High Method Difficulty:**
  - The problem demands a creative or non-routine approach that goes well
     beyond standard adaptations.
  - This includes devising new frameworks, integrating multiple disparate
     methods in unconventional ways, or reasoning through unfamiliar or
    abstract concepts.
  - *Threshold:* If multiple adaptations, non-linear reasoning, or
    innovative synthesis of several methods is required, classify as High
    .

**Evaluation Steps:**

1. **Examine the Method:**
   - Focus exclusively on the method employed in the solution. Ignore the
      overall biological or technical complexity unless it directly
    impacts the methods execution.

2. **Determine the Degree of Adaptation Required:**
   - **Direct Application (Low):** If the method is used in a textbook,
    unmodified way.
```

```
    - **Minor Adaptations (Medium):** If a standard method is modified or
     augmented by one extra layer of interpretation or integration.
    - **Significant Innovation (High):** If the solution requires
     combining multiple methods, developing a new approach, or applying
     the method in a highly non-standard or creative manner.

3. **Provide a Detailed Rationale:**
    - Cite specific aspects of the method in the problem statement and
     reference solution that indicate whether the method is used routinely
      or has been significantly adapted.
    - Explain why any additional steps or integrations push the
     classification toward Medium or High.

4. **Maintain Objectivity:**
    - Base your classification solely on the nature and execution of the
     methodnot on the underlying scientific problem.
    - Emphasize the number and significance of modifications required to
     apply the method.

==================================================
Problem Statement:
<problem_statement>
{problem_statement}
</problem_statement>

==================================================
Reference Solution:
<reference_solution>
{reference_solution if reference_solution is not None else  None }
</reference_solution>

==================================================
Output your answer as a JSON object with the keys  method_difficulty  (
    one of  Low ,  Medium ,  High ) and  rationale . Do not include any
    additional commentary.
```

PROMPT 3: DEFINITION COMPLETENESS ASSESSMENT

```
You are an expert in evaluating scientific problems using Bloom's 3D
    Taxonomy.
Please analyze the following problem statement and reference solution,
    focusing exclusively on whether the problem is completely defined.
Consider the following:
- Is the problem completely defined with all necessary details provided?
    If yes, answer  yes ; if it requires assumptions or strategic
    decisions, answer  no .
Provide a detailed rationale by citing specific aspects of the problem
    statement and reference solution.

==================================================
Problem Statement:
<problem_statement>
{problem_statement}
</problem_statement>

==================================================
Reference Solution:
<reference_solution>
{reference_solution if reference_solution is not None else  None }
</reference_solution>
```

```
Output your answer as a JSON object with two keys  completely_defined  (
     yes  or  no ) and  rationale . Do not include any additional
     commentary.
```

PROMPT 4: KNOWLEDGE DIMENSION ASSESSMENT

```
You are an expert in evaluating scientific problems using Bloom's 3D
    Taxonomy.
Please analyze the following problem statement and reference solution,
    focusing exclusively on the type of knowledge required.
The knowledge dimensions (from lowest level to highest level) are defined
     as:
-  Factual : Involves recalling basic facts or definitions.
-  Conceptual : Involves understanding theories, principles, and
    relationships.
-  Procedural : Involves knowing how to perform methods or algorithms.
-  Metacognitive : Involves awareness and control of o n e s  own thinking
     processes.

Your tasks:
1. Determine the **highest level** of knowledge dimension(s) the problem
    targets.
2. Provide a detailed rationale for your classification by citing
    specific elements from the problem statement (and reference solution,
     if provided).

Problem Statement:
<problem_statement>
{problem_statement}
</problem_statement>

Reference Solution:
<reference_solution>
{reference_solution if reference_solution is not None else  None }
</reference_solution>

Please output your answer as a JSON object with two keys:
     knowledge_dimension  (highest level of  Factual ,  Conceptual ,
     Procedural ,  Metacognitive  required) and  rationale . Do not
     include any additional commentary.
```

PROMPT 5: KNOWLEDGE BREADTH ASSESSMENT

```
You are an expert in evaluating scientific problems using an extended
    version of Bloom's Taxonomy.
Focus exclusively on the 'Knowledge: Breadth' subdimension. This
    dimension examines whether the problem
requires integrating knowledge from multiple disciplines.
By order of complexity, from low to high, this dimension's levels are
    defined as:
-  Single-Discipline : Does not require integrating knowledge from
    multiple disciplines.
-  Multi-Discipline : Requires integrating knowledge from multiple
    disciplines.

Your tasks:
1. Analyze the problem statement and reference solution.
2. Determine if the problem involves multiple disciplines (i.e.,  Multi-
    Discipline ) or is confined to one.
3. Provide a detailed rationale citing specific elements from the text.
```

```
Output your answer as a JSON object with two keys:  knowledge_breadth
    and  rationale . Do not include any additional commentary.

Problem Statement:
{problem_statement}

Reference Solution:
{reference_solution if reference_solution is not None else  None }
```

PROMPT 6: REASONING STEPS ASSESSMENT

```
You are an expert in evaluating scientific problems using an extended
    version of Bloom's Taxonomy.
Focus exclusively on the 'Reasoning: Number of Reasoning Steps Required
    to Solve the Problem' subdimension. This dimension examines how many
    distinct
reasoning or planning steps are required to solve the problem, and
    whether the process is straightforward or complex.

Follow the guidelines below to count the number of distinct reasoning
    steps:
1. **Identify Independent Logical Actions:**
   - Count each separate act of analysis, deduction, computation, or
   planning that contributes uniquely to arriving at the solution.
   - A \ s t e p  should represent an independent logical move rather than
   a mere rephrasing or elaboration of an earlier step.

2. **Define the Boundaries of a Step:**
   - A step begins when a new reasoning method or operation is introduced
     (e.g., setting up an equation, deducing a chemical property, or
   applying a theoretical principle).
   - A step ends when that piece of reasoning has been fully applied or
   resolved. Avoid splitting actions that are inherently part of one
   unified idea.

3. **Ensure Each Step Is Essential:**
   - Only count steps that are necessary for reaching the final
   conclusion. Trivial clarifications, restatements of known facts, or
   background information that does not directly contribute to the
   reasoning process should not be counted as separate steps.
   - Consider whether the removal of a step would leave a gap in the
   logical progression toward the solution. If so, it must be counted.

4. **Differentiate Between Substeps and Major Steps:**
   - When a step naturally divides into substeps, assess if those
   substeps represent distinct reasoning actions. If they are tightly
   interwoven and the separation does not change the logical flow, count
    them as one step.
   - Use clear criteria such as \new calculation , \new inference ,
   or \application of a different p r i n c i p l e  to decide on splitting or
    merging substeps.

5. **Apply Consistent and Objective Criteria:**
   - Use objective markers such as \setting up an e q u a t i o n , \balancing
    mass or c h a r g e , \inferring molecular s t r u c t u r e , \evaluating
   experimental d a t a , or \applying a t h e o r y  to identify steps.
   - Ensure that the criteria for what counts as a step are uniformly
   applied, regardless of the scientific discipline (chemistry, physics,
    biology, etc.).

6. **Document the Rationale for Each Counted Step:**
```

```
   - For every reasoning step counted, provide a brief description that
    clearly indicates why it is independent and essential to the overall
    solution.
   - This documentation should include reference to the specific part of
    the problem or reasoning process it addresses.

7. **Sequence and Dependency Consideration:**
   - Ensure that each counted step represents a sequential action that
    builds on previous steps. Independent parallel reasoning that
    converges into one conclusion may be considered as separate steps if
    each contributes a distinct part to the final result.
   - Avoid double-counting: if two pieces of reasoning essentially
    support the same conclusion without introducing new independent
    information, they should be merged into one step.

=================================================
Problem Statement:
<problem_statement>
{problem_statement}
</problem_statement>

=================================================
Reference Solution:
<reference_solution>
{reference_solution}
</reference_solution>

=================================================

Your tasks:
1. Analyze the problem statement and reference solution.
2. Output the number of distinct reasoning or planning steps required to
    solve the problem.
3. Provide a detailed rationale citing specific elements from the text.

Output your answer as a JSON object with two keys:  number_of_steps  and
    rationale . Do not include any additional commentary.
```

These prompt templates are designed to maximize annotation consistency and interpretability. For further implementation details, see Section 2.2.

Bloom Distribution Plots on GPQA, SuperGPQA and ARC To analyze how scientific problems are distributed across the Bloom taxonomy, we visualize the annotation results across three benchmarks: GPQA, SuperGPQA, and ARC. These distributions reveal the relative frequency of cognitive, knowledge-level, method difficulty demands placed by each benchmark.

### 7.2.1 HUMAN ALIGNMENT FOR BLOOM METRICS

### 7.3 PLOTS FOR REASONING EFFORT METRICS

We visualize the distribution of various reasoning metrics across models. These include:

- Wrong Answer Fraction (WAF) — Figure 12

- Minimum Reasoning Tokens for Correct Answer — Figure 13

- Expected Runs to Correctness — Figure 14

- Uncertainty in Correct Answers — Figure 15

- Inter-run Reasoning Inconsistency — Figure 16

These metrics serve as quantitative signals of reasoning effort and complexity in LLM behavior, complementing the structural annotations from the Bloom taxonomy.

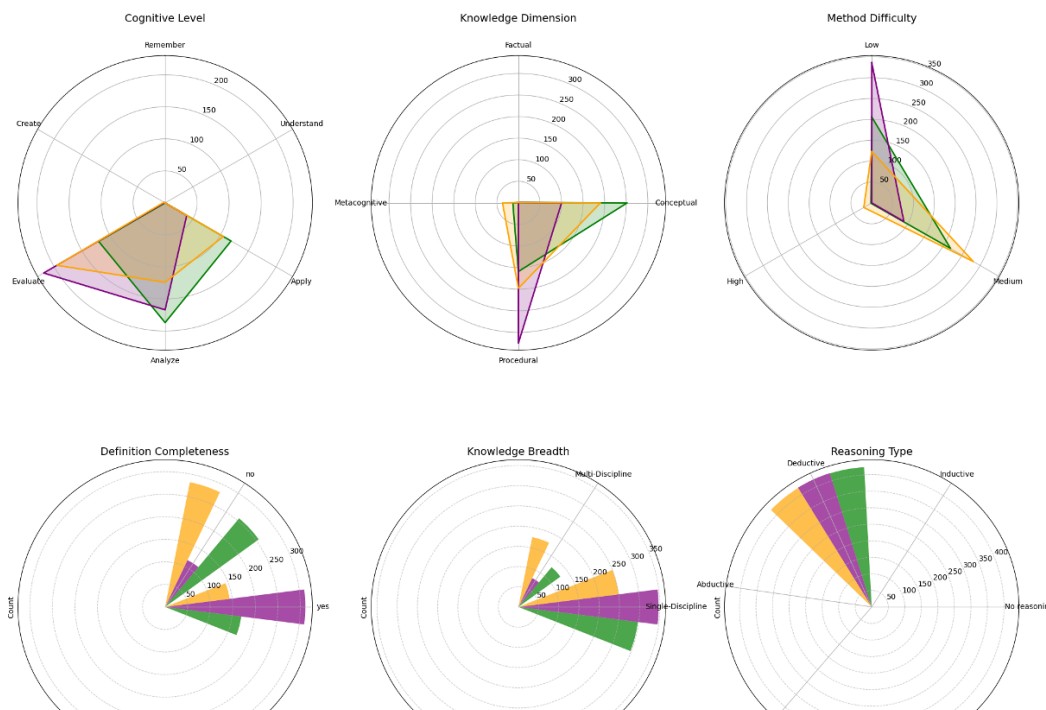

Figure 8: Bloom Metrics Distribution for GPQA

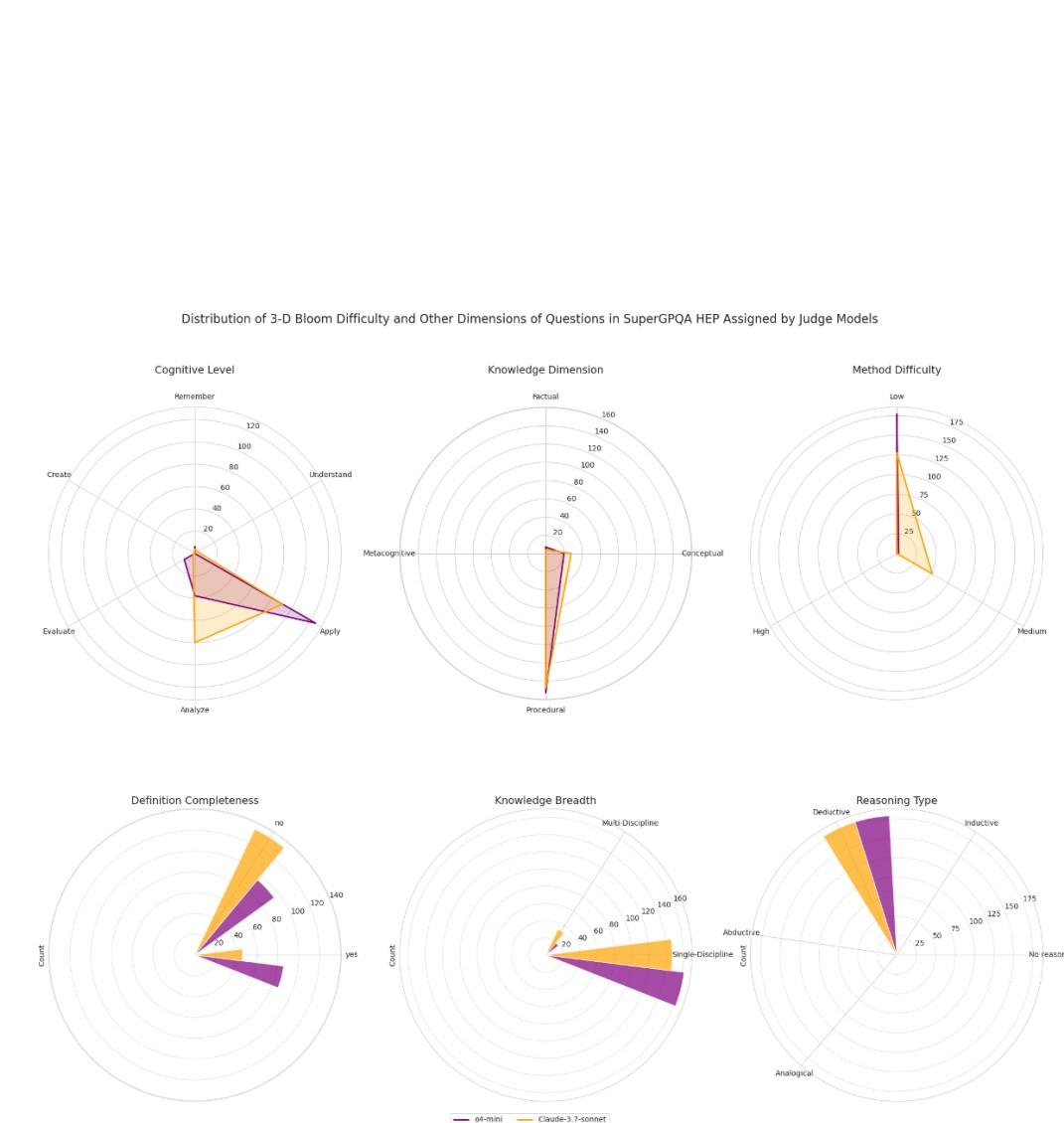

Figure 9: Bloom Metrics Distribution for SuperGPQA

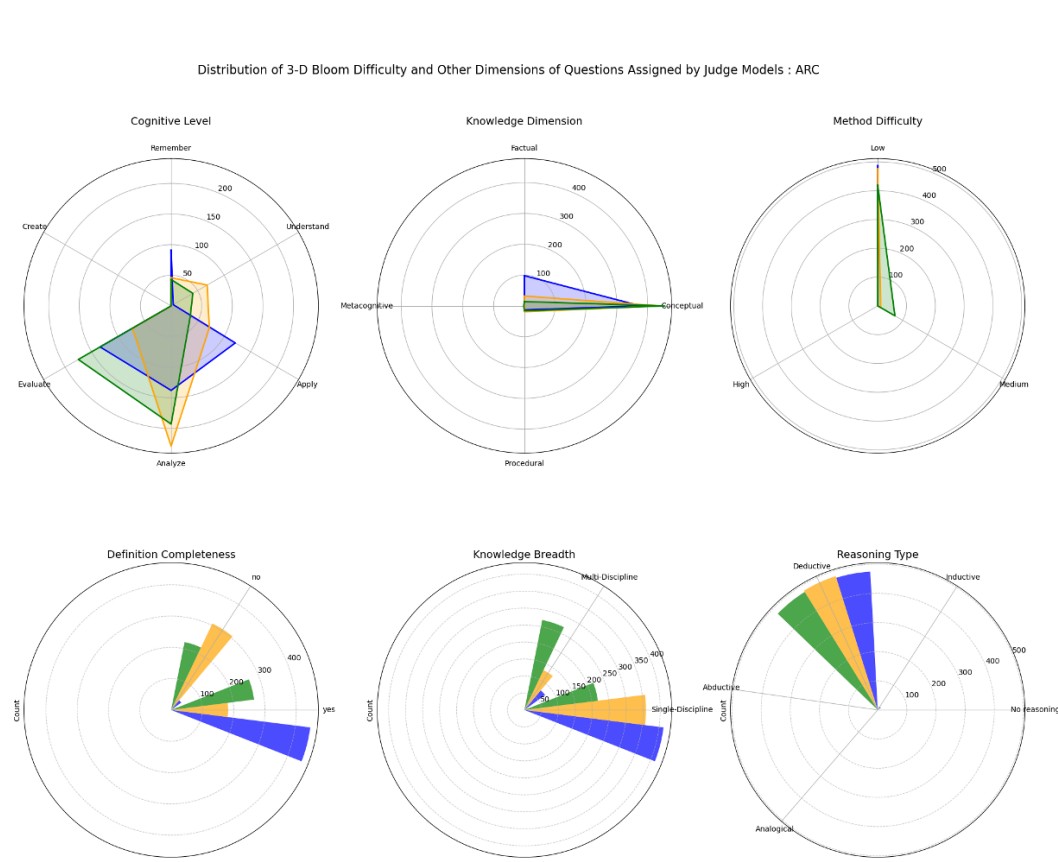

Figure 10: Bloom Metrics Distribution for ARC

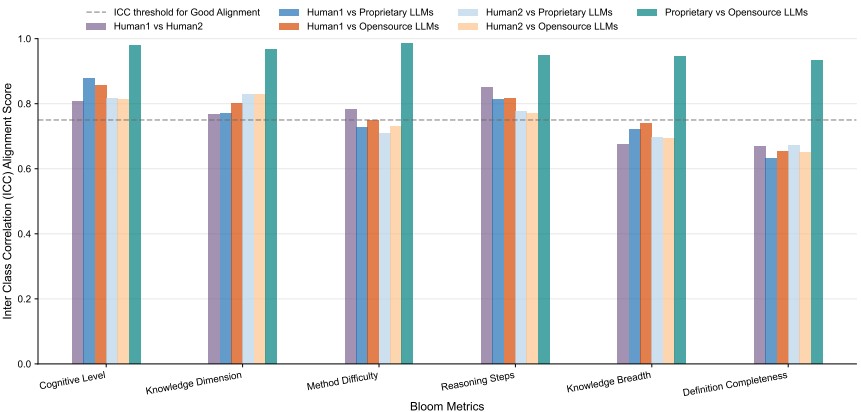

Figure 11: Human Alignments and LLM alignment

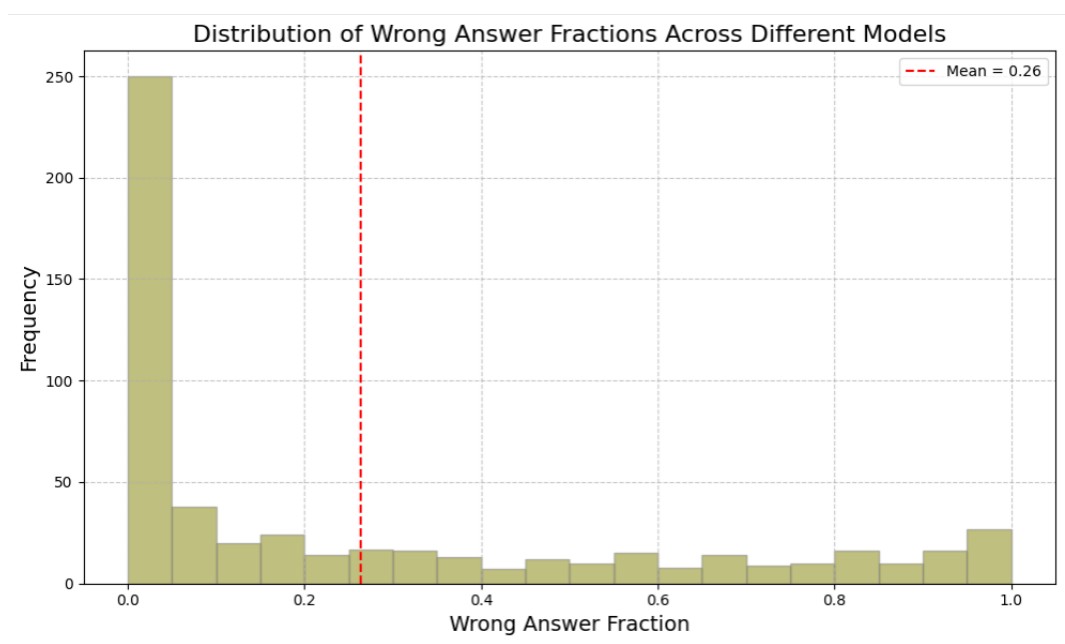

Figure 12: Distribution of Wrong Answer Fraction(WAF)

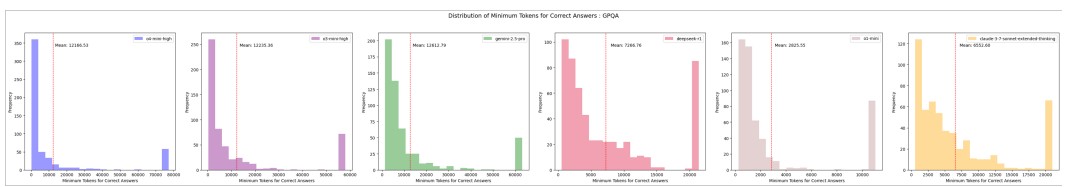

Figure 13: Distribution of Minimum Reasoning Token for Right Answer across Models $M$

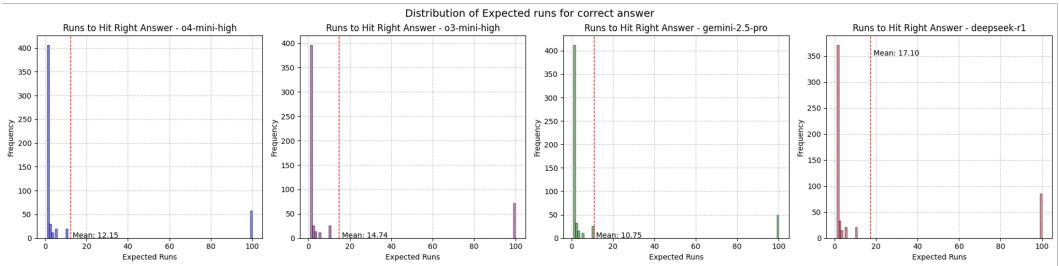

Figure 14: Distribution of Expected Runs for Correct Answer

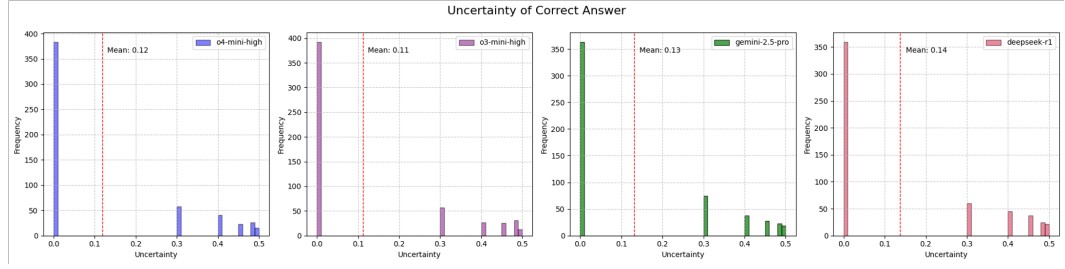

Figure 15: Distribution of Uncertainty of Correct Answers

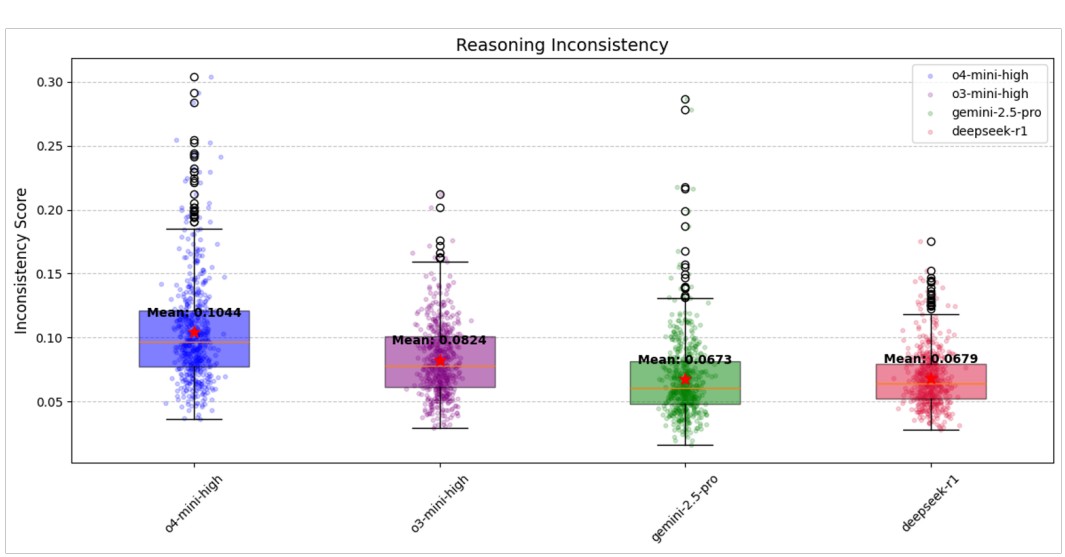

Figure 16: Distribution of Reasoning Inconsistency across 10 runs

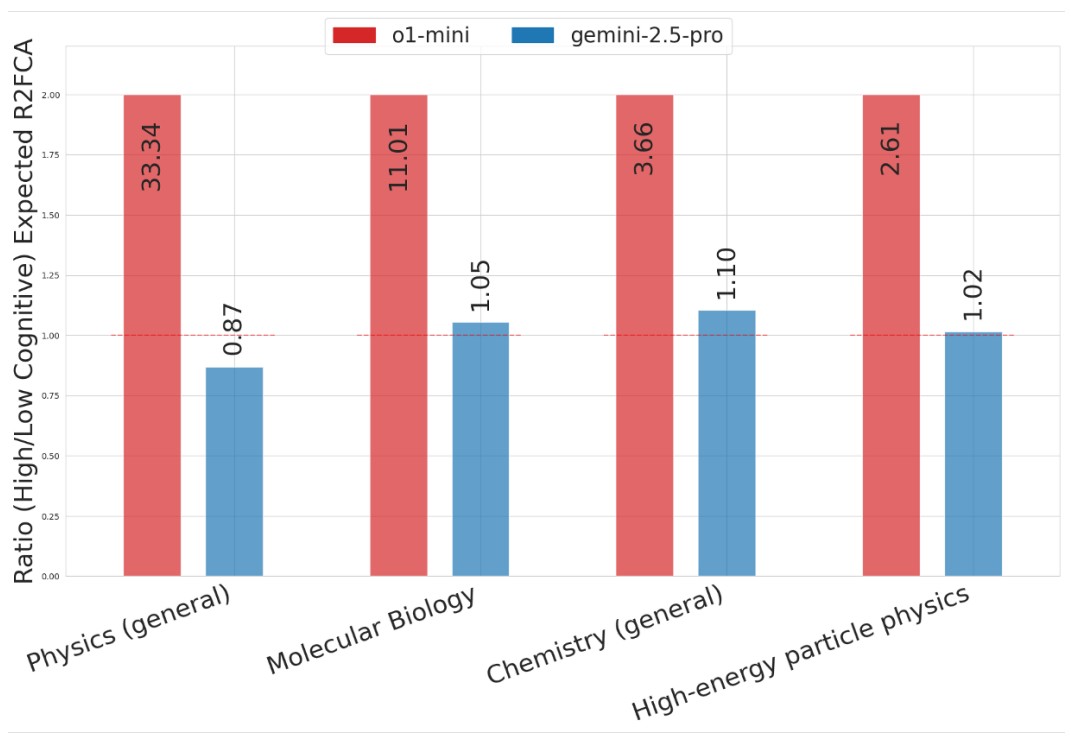

Figure 17: Ratio of High to Low Cognitive Expected R2FCA for o1-mini and Gemini-2.5-pro across 4 GPQA subfields

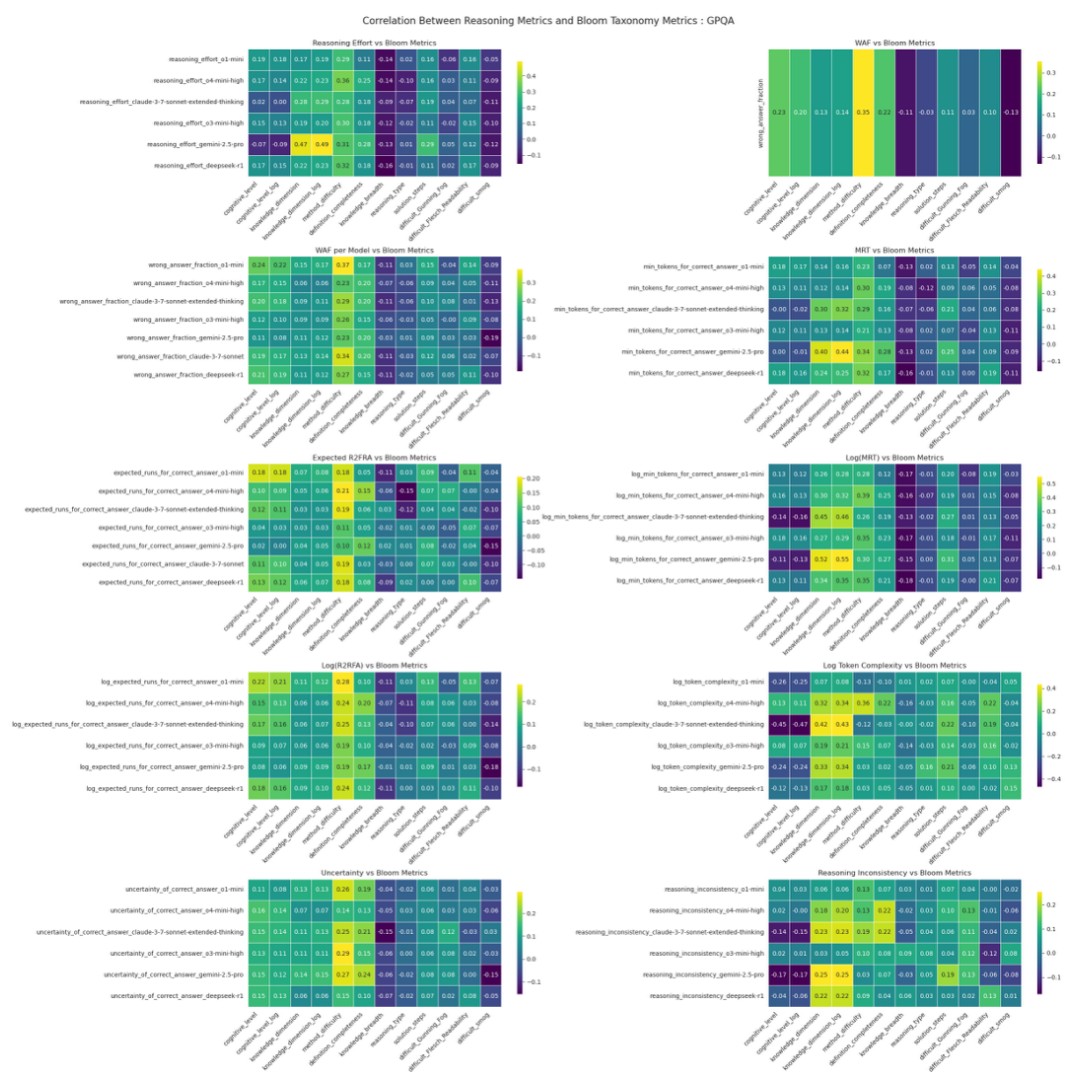

Figure 18: Correlation of Bloom Metrics across Reasoning Effort Metrics from individual reasoning models: GPQA

## 7.4 EXTENDED CORRELATION ANALYSIS

For ARC we also leveraged the human labeled data provided by Clark et al. (2018) to perform correlation analysis to see if the human labeled metrics align well with the judge models Bloom prediction. The human labeled score was only for cognitive levels, it does not contain other dimensions of bloom taxonomy. We see that the human bloom scores correlate equally well as model assigned bloom scores across reasoning metrics.

## 7.5 MODEL GENERATION COMPARISONS ON REASONING EFFORT

We compare different model generations—such as `o1-mini`, `o3-mini`, and `o4-mini`—with respect to the reasoning effort required for problems of varying difficulty:

- Figure 22 compares minimum reasoning token lengths across knowledge difficulty classes.
- Figure 23 summarizes model behavior across effort metrics.

These comparisons illustrate emergent capabilities in later model generations, especially in handling high-difficulty tasks.

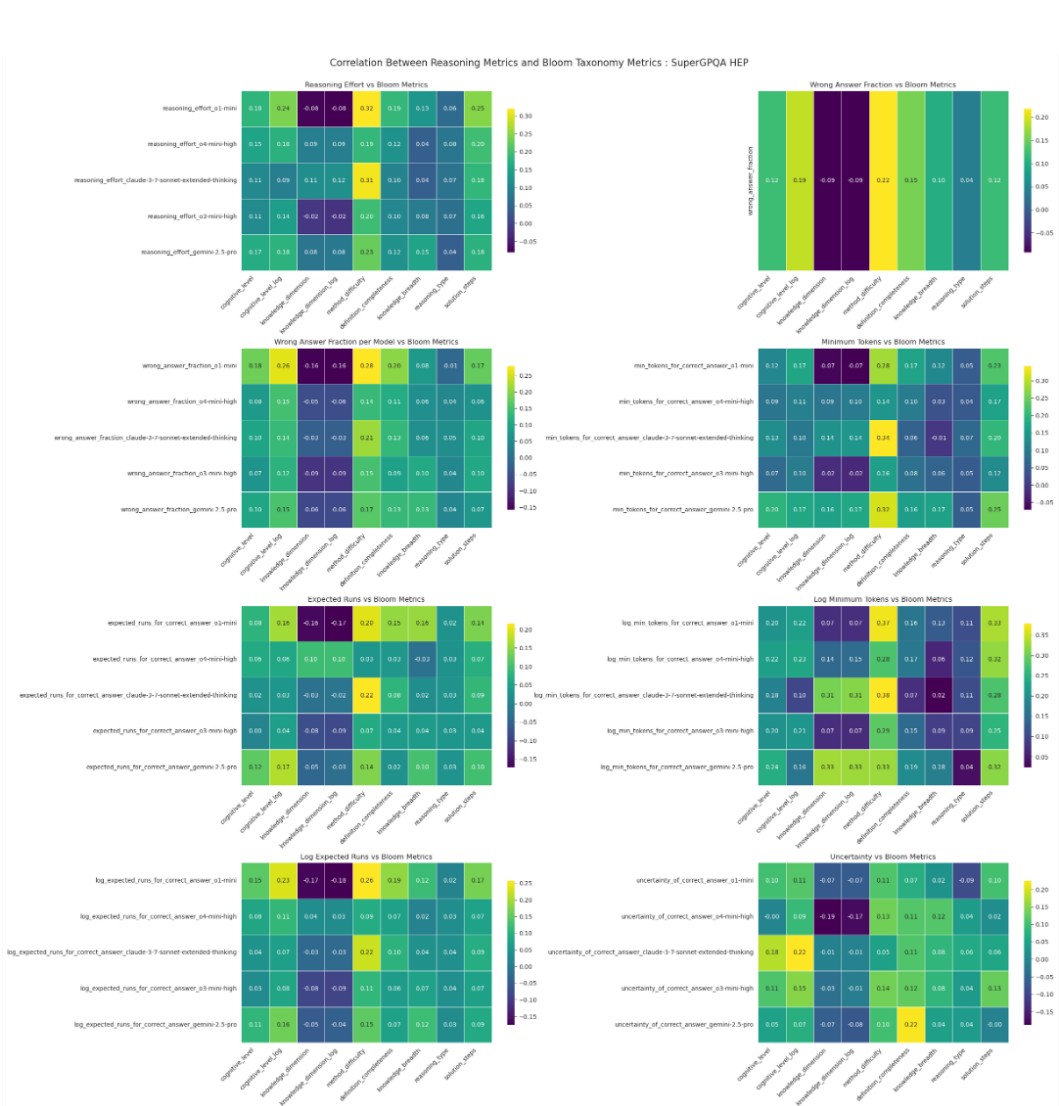

Figure 19: Correlation of Bloom Metrics across Reasoning Effort Metrics from individual reasoning models : SuperGPQA

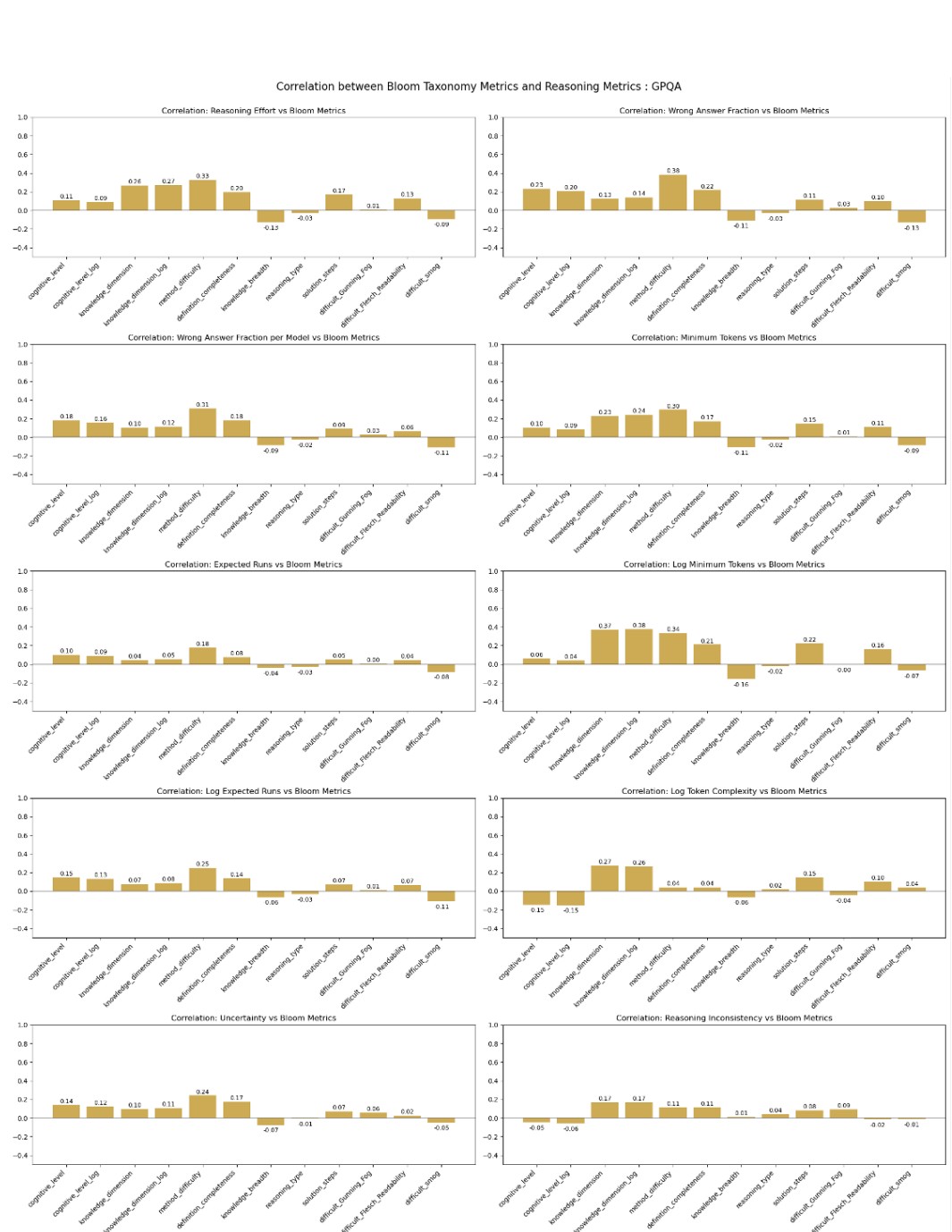

Figure 20: Overall correlation of Individual Reasoning Metrics w.r.t Bloom metrics

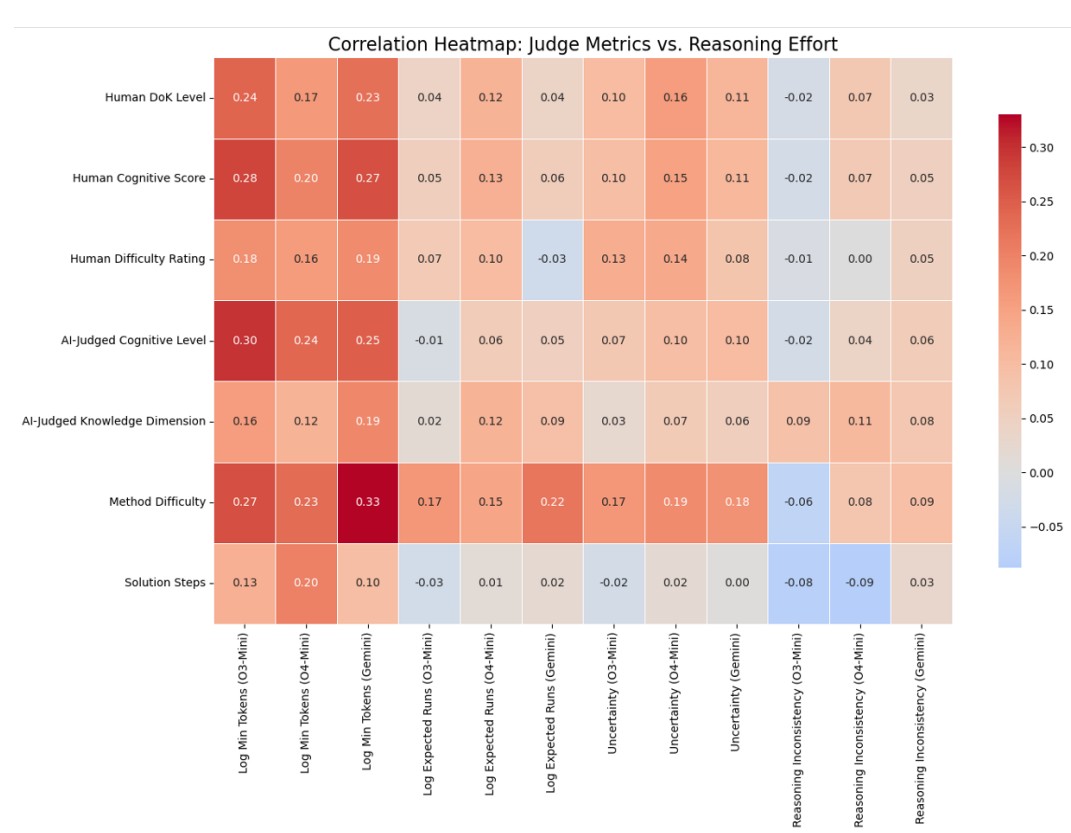

Figure 21: Correlation of Bloom Metrics across Reasoning Effort and Human Labeled Difficulty : ARC

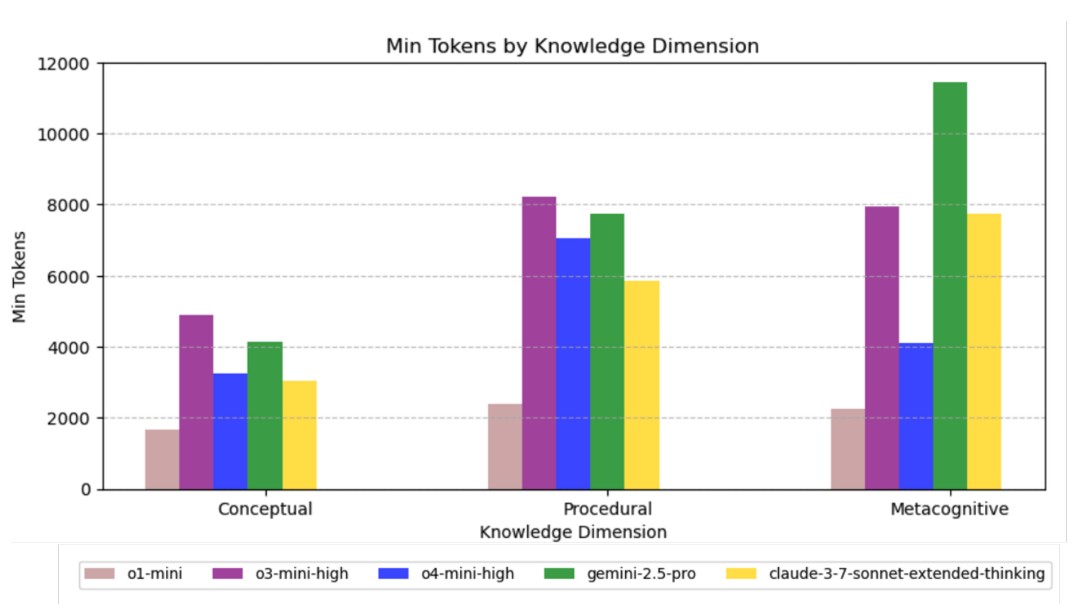

Figure 22: Minimum Reasoning Tokens to Right Answer across Knowledge Difficulty

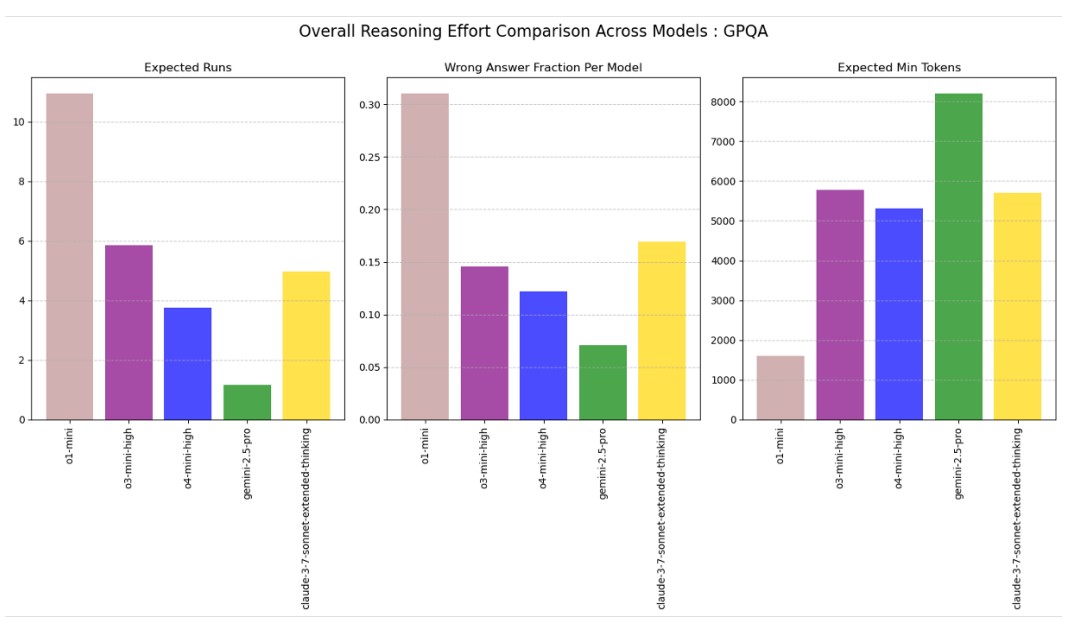

Figure 23: Overall analysis across Models

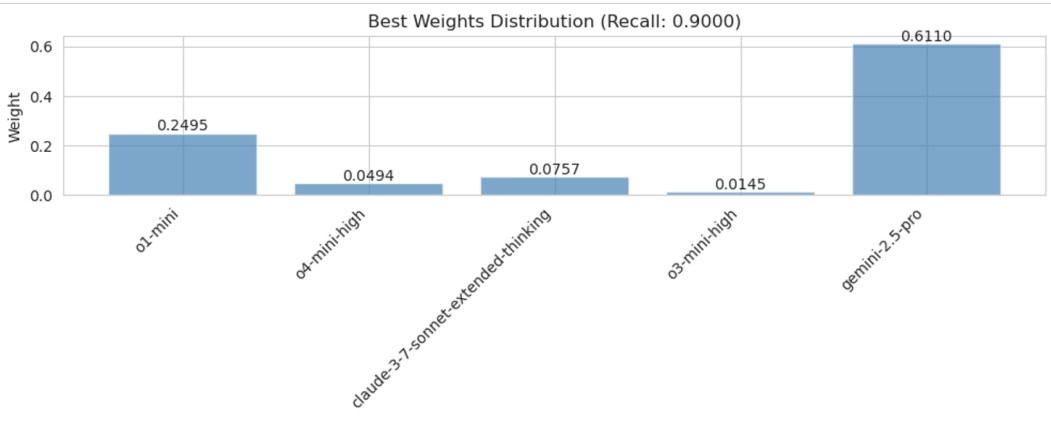

Figure 24: Weights Optimised for Recall : GPQA

## 7.6 REASONING EFFORT PREDICTION

Finally, we present results from Combined Reasoning effort model that predict reasoning effort from Bloom taxonomy features. Key findings include:

- Figure 24 and Figure 25 show weights learned for recall and F1-optimized classification, respectively.
- Figure 26 illustrates classifier performance for detecting high-effort reasoning cases optimized through f1.

### 7.6.1 PROMPT FOR LLM RUBIC CLASSIFICATION

Using a Judge LLM, in our experiment we have used o4-mini-high, to perform reasoning effort classification to serve as a baseline for DoReMi approach. The Judge LLM is given the problem statement, and reference solution as context to classify the reasoning effort required to solve the problem.

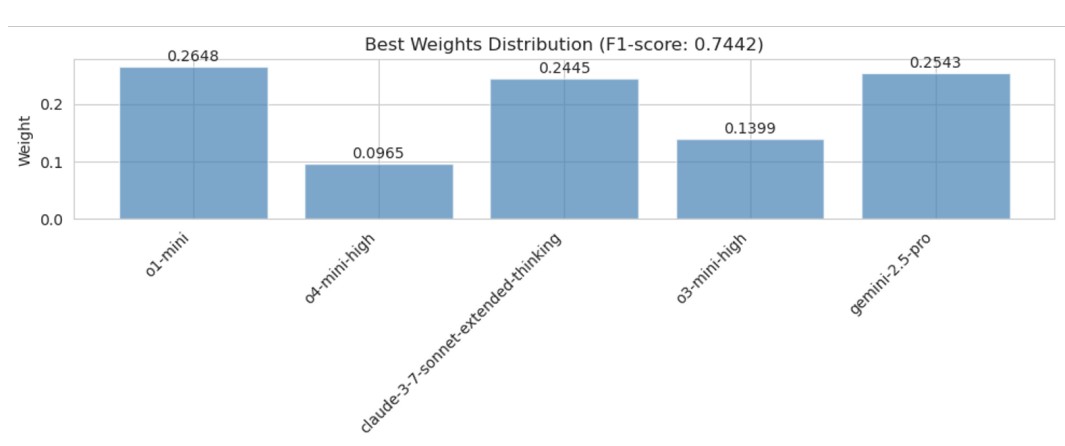

Figure 25: Weights Optimized for F1-score: GPQA

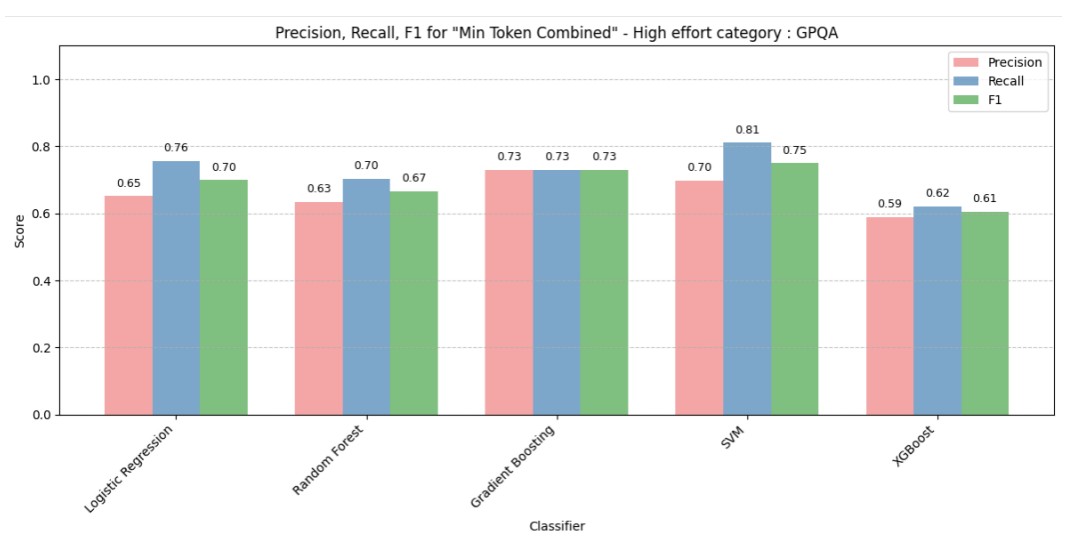

Figure 26: Model Prediction Metrics for Classifying High Reasoning Effort

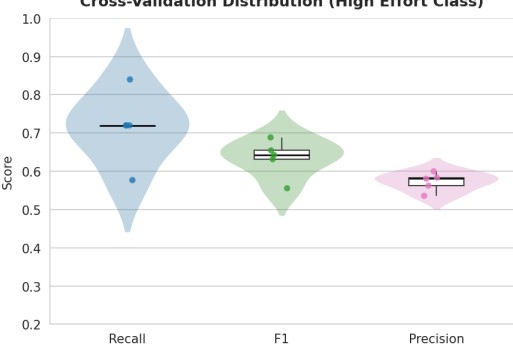

Figure 27: Confidence interval for predicting High Reasoning Effort class

```
You are a top-notch scientist. Classify the reasoning effort required to
    solve the given scientific problem into exactly one category: Minimum
    , Low, Medium, or High.

## Definitions

**Minimum:** Direct recall or single-step application. Problem is solved
    by retrieving and directly applying one known fact, formula, or
    procedure.

**Low:** Straightforward multi-step reasoning within a single concept.
    Require 2-4 logical steps using one domain of knowledge, with minimal
     abstraction or transformation.

**Medium:** Coordinated application of multiple concepts. Requires
    selecting appropriate methods, combining knowledge from 1-2 domains,
    or building intermediate representations to bridge problem and
    solution.

**High:** Complex integration across domains. Demands synthesizing
    concepts from 3+ domains, constructing elaborate models, navigating
    significant abstraction, or developing novel solution pathways.
    Sometimes require a large number of reasoning steps to solve the
    problem.

## Guidelines

    - Consider the conceptual complexity, not computational difficulty
    - If you consider the reasoning steps, use the cognitive steps an
    expert would perform, not the time required
    - Provide a rationale (2-3 sentences) that identifies the key
    reasoning operations and justifies your classification

## Problem Statement:
{problem_statement}

## Reference Solution:
{reference_solution if reference_solution is not None else  None }
```