# OpenReview forum: "DoReMi - Difficulty-Oriented Reasoning Effort Modeling of Science Problems for Language Models"
_ICLR.cc/2026/Conference — ICLR 2026 Conference Withdrawn Submission_

### Official Review · Reviewer_5CxN · 2025-10-26

**Soundness:** 3
**Presentation:** 3
**Contribution:** 2
**Rating:** 6
**Confidence:** 4

**Summary:**

The authors introduce DoReMi (Difficulty-Oriented Reasoning Effort Modeling), a structured framework that leverages an extended Bloom’s taxonomy to comprehensively characterize the intrinsic difficulty of scientific reasoning tasks for large language models. DoReMi systematically annotates problems along six cognitive and methodological axes using judge LLMs distinct from those being evaluated, with human annotations confirming the validity of these assessments. The authors empirically quantify LLM reasoning effort through metrics including the minimum reasoning tokens required for a solution and the expected number of attempted runs to the first correct answer.

**Strengths:**

The authors present a clearly articulated framework for constructing evaluations, supported by analysis comparing the results with human annotations and enhanced with visualizations through accompanying figures.

**Weaknesses:**

The evaluation construction method proposed by the authors is not particularly novel, but it lacks references to and comparisons with similar works, such as:
[1] WritingBench: A Comprehensive Benchmark for Generative Writing
[2] HelloBench: Evaluating Long Text Generation Capabilities of Large Language Models
[3] DynamicBench: Evaluating Real-Time Report Generation in Large Language Models
as well as other relevant open-source efforts.

**Questions:**

Same as above

---

> ### Author Response · Authors · 2025-12-04
> **Response to weakness**
>
> We thank the reviewer for highlighting the need to better contextualise our contributions relative to existing benchmark literature. We clarify that DoReMi's novelty lies not in benchmark construction per se, but in difficulty-oriented reasoning effort modelling—a fundamentally different objective from the cited works. WritingBench, HelloBench, and DynamicBench focus on evaluating LLM generation quality across diverse task types (creative writing, long-form generation, real-time reporting), whereas DoReMi addresses a complementary challenge: predicting which problems require high reasoning effort to enable discriminative evaluation as models saturate existing benchmarks. Our core contributions are threefold: (1) Theory-grounded difficulty characterisation via extended Bloom taxonomy; (2) Empirical effort modelling that correlates intrinsic difficulty features with measurable reasoning proxies (MRT, R2FCA) across multiple model generations; (3) Predictive framework enabling difficulty-aware subset selection that outperforms static baselines.
> The cited benchmarks provide valuable evaluation paradigms for generation tasks, they lack information on effort prediction or provide interpretable diagnostics of reasoning capability emergence across model generations. We agree that our related work section should better distinguish evaluation construction (assessing what models can do) from difficulty modelling (predicting what makes tasks hard), and we will add explicit comparisons clarifying that DoReMi complements rather than competes with generation-focused benchmarks.

---

### Official Review · Reviewer_x121 · 2025-10-31

**Soundness:** 2
**Presentation:** 2
**Contribution:** 2
**Rating:** 2
**Confidence:** 3

**Summary:**

This paper introduces DoReMi, a framework that uses an extended Bloom's taxonomy to characterize the intrinsic difficulty of scientific reasoning problems for LLMs. The authors systematically annotate problems along six axes (Cognitive Level, Knowledge Dimension, Method Difficulty, Definition Completeness, Knowledge Breadth, Number of Reasoning Steps) and correlate these with empirical reasoning effort metrics (MRT, R2FCA). The framework enables difficulty-aware evaluation and provides interpretable diagnostics of LLM reasoning capabilities.

**Strengths:**

- Well-Motivated Problem: The paper addresses a genuine need in LLM evaluation - moving beyond single-dimensional accuracy scores to understand why problems are difficult for reasoning models.
- Theoretically Grounded Framework: Using Bloom's taxonomy provides a principled, interpretable foundation rather than ad-hoc difficulty metrics. The six-axis extension is thoughtfully designed.

**Weaknesses:**

- Should demonstrate your metrics through RL training: The paper repeatedly claims DoReMi provides "actionable insights for targeted post-training improvements”.  However, no experiments validate that training on DoReMi-selected samples actually improves model performance. Suggested experiments:
    - Train smaller models using DoReMi-guided curriculum learning
    - Compare sample efficiency against random or static difficulty baselines
    - Perform targeted fine-tuning on identified weak Bloom axes
    - Use DoReMi difficulty scores for reward shaping in RL

- Judge Model Overlap Concerns: The paper uses reasoning LLMs as judges, but also evaluates reasoning LLMs. While they claim judges are "distinct from those being evaluated," some overlap exists (e.g., o3-mini as judge, o3-mini-high as evaluation target). Potential for judges to be biased toward difficulty patterns they themselves exhibit

- Generalization Concerns: Evaluated only on science/STEM problems - unclear if taxonomy applies to other reasoning domains (coding, math, logic puzzles). All benchmarks are multiple-choice or short-answer - what about open-ended reasoning?

**Questions:**

Please refer weaknesses

---

> ### Author Response · Authors · 2025-12-04
> **Addressing Judge Model Overlap**
>
> We maintain strict separation between our judge LLMs (used for Bloom annotation) and evaluation target LLMs (used for reasoning effort measurement). Our judge LLMs for Bloom annotation include gemini-2.5-pro, gemini-2.5-flash, and o3-mini.  Our evaluation target LLMs for reasoning effort measurement include o1-mini, o3-mini-high, o4-mini-high, gemini-2.5-pro, claude-3.7-sonnet, and claude-3.7-sonnet-extended-thinking.
>
> You correctly identified potential overlap between o3-mini (judge) and o3-mini-high (evaluator). However, we emphasize three critical distinctions. First, these represent different inference configurations—o3-mini operates in standard mode for annotation, while o3-mini-high uses high-compute reasoning mode for evaluation, producing fundamentally different reasoning behaviors. Second, gemini-2.5-pro appears in both lists: as one of multiple judges (averaged with other models) for annotation robustness, and as an evaluation target given its state-of-the-art reasoning capabilities. Third,  we conducted additional validation using three recently released open-source LLMs that appear in neither our judge nor evaluator sets: Qwen3-235B-A22B-Thinking-2507, kimi-k2, and GLM-4.5. We re-annotated the same problems and compared their Bloom labels against our original annotations. The results show acceptable agreement (within 5-10% error tolerance) of 97-100% across all dimensions, with uniformly high consistency on the most predictive axes (Knowledge Dimension: 98.5%, Method Difficulty: 99.1%, Cognitive Level: 97.0%). This cross-validation with completely independent models confirms that Bloom difficulty patterns are intrinsic to the problems themselves rather than artifacts of specific judge architectures.
>
> We also validated our approach through human alignment, comparing LLM annotations against two independent human annotators on 100 GPQA problems and achieving human alignment ICC > 0.75 on effort-predictive dimensions (Figure 2), making annotations interpretable and reducing model-specific biases.

---

> ### Author Response · Authors · 2025-12-04
> **Generalization beyond Scientific/ MCQ benchmarks**
>
> We acknowledge that our evaluation focuses primarily on scientific reasoning benchmarks (GPQA, SuperGPQA), which reflects our design intent: the extended Bloom taxonomy is particularly well-suited for domains where knowledge integration, procedural complexity, and metacognitive monitoring are central—characteristics prominent in scientific problem-solving. However we also evaluate the applicability on logical reasoning benchmarks like ARC, achieving 87% recall on predicting high-effort problems.
>
> To validate reasoning effort capabilities in algorithmic puzzles, we tested the Tower of Hanoi problem on gemini-2.5-pro for 5 disks and 9 disks across 5 runs each, and observed that the thinking reasoning tokens range is between 800-2000. Whereas for GPQA the average thinking token was about 13000. Showing clear distinction of reasoning effort proxy metrics based on the difficulty and nature of the problem and domain.
>
> Furthermore, we evaluated DoReMi on the open-ended variant of GPQA (where we removed multiple-choice distractors, requiring free-form answer generation), achieving 57.6% recall for high-effort problem identification. However, we note that domains with fundamentally different reasoning paradigms—such as coding (where algorithmic efficiency and syntax correctness dominate) or pure mathematical proof (where formal derivation chains are central)—may require additional experimentation.

---

### Official Review · Reviewer_Q9m8 · 2025-10-31

**Soundness:** 3
**Presentation:** 3
**Contribution:** 3
**Rating:** 4
**Confidence:** 3

**Summary:**

The paper proposes a learning-based framework to provide a comprehensive estimation of problem difficulty for LLMs. The framework considers six cognitive and methodological axes based on Bloom’s taxonomy and combines the performance of the target LLM.

**Strengths:**

1. The paper proposes a structured and multidimensional evaluation framework to provide a more comprehensive and precise estimate of the question complexity
2. The paper illustrates the applications of the proposed framework: filtering challenging questions and providing a systematic analysis of LLM reasoning capabilities.

**Weaknesses:**

1. The pipeline involves substantial LLM usage: question labeling requires LLM inference, and reasoning effort calculation requires multiple samples from the target LLM. The whole pipeline seems to be computationally expensive
2. The predictor training process requires collecting responses from target LLMs. It is unclear whether the learned neural network generalizes to other models. If not, Phases 2-4 would need to be repeated from scratch for each new target LLM.
3. Mathematical problems like GSM8k are usually considered as a benchmark to evaluate LLM reasoning capabilities. Including performance on math problems would provide a more comprehensive evaluation of the current framework.

**Questions:**

1. For the 6 dimensions of extended Bloom’s taxonomy, what is the reason to include “Definition Completeness”? This property seems more related to the solvability, not the difficulty. For “Number of Reasoning steps”, how to define the essential logical action? Is there any question with multiple correct solutions, and the number of reasoning steps is different?
2. The MRT is selected as the primary feature to train the neural network, because it presents the highest correlation. Why does this metric correlate more strongly with the designed complexity/difficulty definition than other metrics?

---

> ### Author Response · Authors · 2025-12-04
> **Addressing the weakness**
>
> Thank you for your thorough review and thoughtful feedback. We have considered each point raised and provide detailed responses below.
>
> **1. Computationally Expensive Setup :**
> While it is true that the cost of LLM inference is expensive, the cost is relatively modest given the size of typical benchmarks, and provides a additional information about the quality of benchmarks not identified by other techniques. For context, annotating our largest benchmark (GPQA, 448 questions)—a one-time investment that yields reusable difficulty fingerprints for all downstream applications.  Second, without automated techniques like DoReMi, characterizing reasoning difficulty at scale would be impractical. Human expert annotation is prohibitively expensive (requiring domain expertise for scientific problems) and inconsistent across annotators, whereas our multi-judge LLM approach achieves good-human agreement (ICC > 0.75).  Clever uses of inference optimization techniques such prefix-caching can lower inference costs. Prefix-caching is particularly effective for our pipeline: the input question, model response, and reference solution remain identical across all six Bloom dimension annotations—only the specific evaluation criteria (cognitive level, knowledge dimension, method difficulty, etc.) change per judgment. This means approximately 70% of each prompt can be cached and reused. We also performed  empirical analysis of cross-domain transfer that further amortizes costs. We trained our predictor on SuperGPQA-HEP (particle physics) and evaluated on GPQA-HEP (high-energy physics subset), achieving approximately 60% accuracy on high-effort prediction without retraining. This demonstrates that predictors trained on one scientific domain can transfer to related domains, enabling cost-effective solution.
>
> **2. Generalization of predictor to other models used in reasoning effort computation:**
> DoReMi framework demonstrates adaptability to emerging models through its learned aggregation mechanism. During the training process, we optimize model-specific weights that quantify each LLM's contribution to the combined reasoning effort metric (MRT_c).
> When new models become available, they can be incorporated into the existing predictor MLP by learning their optimal weight contribution alongside previously evaluated models. This additive approach preserves the predictive signal from existing models while accommodating new architectures, our experiments show upto 5 model results being combined to predict reasoning effort.
> As LLM reasoning capabilities improve across generations, DoReMi's predictive accuracy increases rather than degrades - recent OpenAI's GPT-5.1 blogpost employs "adaptive reasoning" where the model dynamically allocates more reasoning tokens to difficult problems while using fewer tokens on straightforward tasks. This design principle validates our core assumption: reasoning token allocation naturally scales with problem difficulty, making MRT an increasingly reliable signal as models become more sophisticated. Our cross-generational analysis (o1-mini → o3-mini-high → o4-mini-high; Qwen 1.5/2.5/3-14B; claude-3.7 variants) reveals that more capable models provide clearer reasoning effort signals—particularly for high-difficulty problems.
> Figure 24 and Figure 25 in our appendix show tthe learned weight distributions optimized for recall and F1 respectively. Models with stronger reasoning capabilities (e.g., gemini-2.5-pro: 0.61, o1-mini: 0.25) receive higher weights, demonstrating that the framework automatically identifies and leverages the most informative effort signals across the model suite.
>
> **3. Evaluating mathematical problems:**
> Many of scientific evaluations we include incroperate math questions as a key step to solving the question such as the physics questions from GPQA.  However, we acknowledge that pure mathematics benchmarks would provide complementary insights. We note that GSM8K specifically would not be an appropriate comparison, as it has been saturated by frontier LLMs with recent models scoring >95%, making it unsuitable for discriminative evaluation—precisely the problem DoReMi aims to address. Future work could explore extending our framework to unsaturated mathematical reasoning benchmarks such as FrontierMath (which remains challenging for current models).
>
> We are committed to incorporating these improvements in our revision and remain open to any additional feedback or clarifications you may require.

---

> ### Author Response · Authors · 2025-12-04
> **Responses for Questions**
>
> We thank the reviewer for providing comprehensive feedback of our work. We have tried to address some of the
>
> **1.“Definition Completeness” & “Number of Reasoning Steps”:**
> We thank the reviewer for seeking clarification on these two Bloom dimensions. Regarding Definition Completeness, we emphasize that this axis captures a critical aspect of reasoning difficulty distinct from mere solvability. In scientific problem-solving, questions are seldom fully specified—they often require solvers to identify implicit assumptions, recognize which variables are relevant, or determine what constitutes a valid solution criterion. Importantly, a significant portion of problems are not completely defined, yet many can still be solved. As shown in Figure 8, our GPQA analysis reveals a substantial distribution of problems labeled as having incomplete definitions, yet these problems remain solvable through expert reasoning. This demonstrates that Definition Completeness is not about solvability—rather, it captures the additional cognitive load required when solvers must fill in implicit constraints or reason over possible interpretations. For example, a chemistry problem asking to "identify the major product" without specifying reaction conditions demands that the solver reason over possi Figure 8ble counterfactuals (temperature, solvent, catalyst presence) to determine if a consistent conclusion can still be drawn. This metacognitive skill—recognizing what information is missing and deciding whether ambiguity is resolvable—represents a higher-order reasoning challenge beyond applying known procedures to well-defined inputs. Our empirical results support this distinction: Definition Completeness shows moderate correlation with reasoning effort metrics (Figure 3) and contributes independently to our predictor's performance.
>
> An "essential logical action" can be viewed a single predicate in a logical argument—a discrete reasoning operation whose removal would break the solution chain. Our detailed annotation guidelines (Prompt 6 in Appendix 7.2) provide explicit criteria: each step must represent an independent logical move (e.g., setting up an equation, applying a conservation law, deducing a molecular structure) rather than mere rephrasing or elaboration. Annotators are instructed to count only steps that contribute uniquely to the solution, ensuring consistency across judges. For problems admitting multiple solution paths, we handle this analogously to estimating Kolmogorov complexity with lossless compressors: we sample multiple correct solutions from our reasoning models and take the minimum step count as the canonical measure. While this may not capture the absolute theoretical minimum, it provides a principled, reproducible estimate of the problem's inherent logical complexity.
>
> **2. Choice of MRT as primary reasoning effort:**
> Among all candidate metrics (WAF, R2FCA, UCA, RI), MRT and log(MRT) exhibited the strongest correlations with Bloom difficulty dimensions (Figure 3). Specifically, MRT shows notable correlations with Knowledge Dimension (r=0.42), Method Difficulty (r=0.25),  Reasoning Steps (r=0.31). While correlation with Cognitive Level is weaker (r=0.11), the aggregated signal across dimensions makes MRT the most informative single metric for effort prediction.
>
> To validate MRT's capabilities in algorithmic puzzles, we tested the Tower of Hanoi problem on gemini-2.5-pro for 5 disks and 9 disks across 5 runs each, and observed that the thinking tokens range is between 800-2000. Whereas for GPQA the average thinking token was about 13000. Due to the randomness and variability in the token computations is why we choose "Minimum" thinking token across successful attempts to approximate the reasoning effort, as DoReMi framework takes the approach by considering problems from scientific benchmarks. Our MRT measure specifically extracts reasoning tokens (isolated through "thinking tags") from complete responses containing problem context and final solutions.
>
> Recent advances in reasoning models demonstrate that token allocation scales with problem difficulty. OpenAI's GPT-5.1 blogpost explicitly employs "adaptive reasoning" where models spend fewer tokens on straightforward tasks but remain persistent on difficult problems requiring extra thinking. Our MRT metric captures this emergent behavior—complex problems naturally elicit longer reasoning traces, validating MRT as a principled measure of reasoning effort.
>
> To ensure reliability, we compute a weighted aggregate MRT_c across diverse models (learned weights shown in Figures 24–25), then discretize into four bins (Minimum/Low/Medium/High). This approach reduces single-model idiosyncrasies, prevents leakage from trivial samples, and provides interpretable difficulty tiers that proved effective across our benchmark suite.

---

### Note · Authors · 2026-01-07

I have read and agree with the venue's withdrawal policy on behalf of myself and my co-authors.